# Mouse B2 SINE elements function as IFN-inducible enhancers

Isabella Horton[†], Conor J Kelly[†], Adam Dziulko, David M Simpson, Edward B Chuong*

Department of Molecular, Cellular, and Developmental Biology and BioFrontiers Institute, University of Colorado Boulder, Boulder, United States

**Abstract** Regulatory networks underlying innate immunity continually face selective pressures to adapt to new and evolving pathogens. Transposable elements (TEs) can affect immune gene expression as a source of inducible regulatory elements, but the significance of these elements in facilitating evolutionary diversification of innate immunity remains largely unexplored. Here, we investigated the mouse epigenomic response to type II interferon (IFN) signaling and discovered that elements from a subfamily of B2 SINE (B2_Mm2) contain STAT1 binding sites and function as IFN-inducible enhancers. CRISPR deletion experiments in mouse cells demonstrated that a B2_Mm2 element has been co-opted as an enhancer driving IFN-inducible expression of *Dicer1*. The rodent-specific B2 SINE family is highly abundant in the mouse genome and elements have been previously characterized to exhibit promoter, insulator, and non-coding RNA activity. Our work establishes a new role for B2 elements as inducible enhancer elements that influence mouse immunity, and exemplifies how lineage-specific TEs can facilitate evolutionary turnover and divergence of innate immune regulatory networks.

*For correspondence:
edward.chuong@colorado.edu

[†]These authors contributed equally to this work

Competing interest: The authors declare that no competing interests exist.

## Editor's evaluation

This important paper will be of interest to scientists studying evolutionary divergence of immune responses and those studying how transposable elements rewire transcriptional regulatory networks. Using a combination of computational and experimental approaches, this work describes a new class of rodent-specific transposons that can act as enhancers of immune genes in mice.

## Introduction

The cellular innate immune response is the first line of defense against an infection and is initiated by the activation of transcriptional networks that include antiviral and pro-inflammatory genes. While innate immune signaling pathways are generally conserved across mammalian species, the specific transcriptional networks are increasingly recognized to show differences across lineages (*Chuong et al., 2017*; *Shaw et al., 2017*). These differences are widely attributed to independent evolutionary histories and continual selective pressures to adapt to new pathogens (*Daugherty and Malik, 2012*). Understanding how innate immune systems have evolved in different host genomes is critical for accurately characterizing and modeling responses that are related to autoimmunity or involved in disease susceptibility.

Transposable elements (TEs) are increasingly recognized as a source of genetic elements that shape the evolution of mammalian innate immune responses (*Chuong et al., 2017*; *Cordaux and Batzer, 2009*). TE-derived sequences constitute roughly half of the genome content of most mammals, and are the predominant source of lineage-specific DNA. While most TE-derived sequences are degraded and presumed nonfunctional, TEs have occasionally been co-opted to function as genes or regulatory

elements that benefit the host organism. In the context of host innate immunity, there are several reported examples of species-specific restriction factors that are encoded by TEs co-opted for host defense, including Friend Virus 1, Syncytin, Suppressyn, and Jaagsiekte sheep retrovirus (JSRV) (*Arnaud et al., 2008*; *Aswad and Katzourakis, 2012*; *Frank et al., 2022*; *Lavialle et al., 2013*). In many cases, TEs derived from ancient viral infections are poised for co-option since they already have the ability to bind to receptors, therefore blocking infection as a dominant negative mechanism (*Frank and Feschotte, 2017*).

More recently, TEs have also been identified as a source of non-coding regulatory elements that control inducible expression of cellular innate immunity genes (*Buttler and Chuong, 2022*). In the human genome, we previously showed that MER41 elements have been co-opted as enhancer elements to regulate multiple immune genes in human cells, including the *AIM2* inflammasome genes (*Chuong et al., 2016*). Elements belonging to other transposon families, including LTR12, MER44, and THE1C, have also been co-opted to regulate inducible expression of immune genes (*Bogdan et al., 2020*; *Donnard et al., 2018*; *Srinivasachar Badarinarayan et al., 2020*). Notably, the majority of these families are primate-specific, supporting the co-option of TEs as a driver of primate-specific divergence of immune regulatory networks.

A key open question is whether the co-option of TEs as immune regulatory elements is evolutionarily widespread as a mechanism driving divergence of innate immune responses. Most research in this area has focused on human cells and primate-specific TE families, but different mammalian species harbor highly distinct and lineage-specific repertoires of TEs in their genomes. Due to the independent origin of most of these TEs in different species, it remains unclear whether the co-option of TEs is a rare or common mechanism contributing to the evolution of immune gene regulatory networks.

Here, we focused on the role of TEs in regulating murine innate immune responses. Mice are a commonly used model for human diseases but their immune system is appreciated to have significant differences. Transcriptomic studies have revealed that mouse and human immune transcriptomes show substantial divergence (*Shaw et al., 2017*; *Shay et al., 2013*), consistent with functional differences in inflammatory responses (*Seok et al., 2013*). The rodent and primate lineages diverged roughly 90 million years ago (*Hedges et al., 2006*; *Mestas and Hughes, 2004*) and 32% of the mouse genome consists of rodent-specific repeats (*Waterston et al., 2002*). Therefore, we sought to define the potential role of TEs in shaping lineage-specific features of the murine innate immune response.

In our study, we re-analyzed transcriptomic and epigenomic datasets profiling the type II interferon (IFN) response in primary mouse macrophage cells. We screened for TEs showing epigenetic signatures of inducible regulatory activity, and identified a rodent-specific B2 subfamily as a substantial source of IFN-inducible regulatory elements in the mouse genome. As a case example, we used CRISPR to characterize a B2-derived IFN-inducible enhancer that regulates mouse *Dicer1*. These findings uncover a novel cis-regulatory role for the SINE B2 element in shaping the evolution of mouse-specific IFN responses.

## Results

### Species-specific TEs shape the epigenomic response to type II IFN in mouse

To examine how TEs contribute to mouse type II IFN signaling regulation, we re-analyzed two independent transcriptomic and epigenomic datasets of primary bone-marrow-derived macrophages (BMDMs) that were stimulated with recombinant interferon gamma (IFNG) or untreated for 2 or 4 hr (*Platanitis et al., 2019*). These datasets included matched RNA-seq and ChIP-seq for STAT1 and H3K27ac. The STAT1 transcription factor mediates the type II IFN response (*Platanias, 2005*) by binding to enhancers and promoters containing the Gamma-IFN activation site (GAS) motif, and the H3K27ac modification is strongly associated with active enhancers (*Creyghton et al., 2010*). Using these datasets, we mapped both IFNG-inducible enhancers and IFNG-stimulated genes (ISGs).

Our analysis of the RNA-seq data identified a total of 1,896 ISGs (FDR adjusted p-value <0.05, $\log_2$ fold change ($\log_2$FC)>1), which enriched for canonical genes associated with the IFNG response (GO:0034341, adjusted p-value = $9.394 \times 10^{-35}$; *Supplementary file 1*). We predicted IFNG-inducible enhancers based on occupancy by the enhancer-associated histone mark H3K27ac and the transcription factor STAT1, which mediates type II IFN signaling. We identified 22,921 regions bound by STAT1

in IFNG-induced cells, 18,337 (80.0%) of which also resided within H3K27ac-enriched regions, indicating they are putative enhancers. Specificity of pulldown was confirmed by enrichment of canonical STAT1 binding motifs including the Gamma-IFN activation site (GAS; E-value=$1.11 \times 10^{-746}$) and IFN-stimulated response element (ISRE; E-value=$1.01 \times 10^{-442}$) motifs within the STAT1 ChIP-seq peaks (*Supplementary file 2*).

Using this set of STAT1 binding sites, we next asked what fraction of binding sites were derived from mouse TEs. Using the summits of the STAT1 ChIP-seq peaks, we found that 26.6% resided within TEs, 71.1% of which contain significant matches (p-value $<1 \times 10^{-4}$) to either ISRE or GAS motifs (*Supplementary file 3*). These TEs likely represent direct binding sites of STAT1 with potential regulatory activity. We next asked whether any TE families were overrepresented within the set of predicted IFNG-inducible binding sites, using GIGGLE colocalization analysis (*Layer et al., 2018*). We identified three subfamilies enriched for STAT1 binding sites, including the rodent-specific B2_Mm2 subfamily (p-value = $7.18 \times 10^{-201}$) as well as the RLTR30B_MM (p-value = $9.61 \times 10^{-77}$) and RLTR30E_MM (p-value = $7.32 \times 10^{-31}$) endogenous retrovirus subfamilies (*Figure 1A*, *Supplementary file 4*). This indicates that the expansion of rodent-specific TE families has shaped the innate immune regulatory landscape in mouse.

We previously identified enrichment of RLTR30 elements within STAT1-binding sites in IFNG and IFNB-stimulated mouse macrophages based on analysis of a different ChIP-seq dataset (*Chuong et al., 2016*; *Ng et al., 2011*). However, our previous analysis did not capture enrichment of B2_Mm2, likely because the dataset was generated using 36 bp short reads. In contrast, the more recent datasets analyzed here used 50 bp reads (*Platanitis et al., 2019*), which improves mappability to individual copies of evolutionarily young TE families such as B2_Mm2 (*Sundaram et al., 2014*).

## B2_Mm2 elements contain STAT1 binding sites and show inducible enhancer activity

B2_Mm2 is a murine-specific subfamily of the B2 short interspersed nuclear element (SINE) family, which is highly abundant in the mouse genome. B2 SINE elements are divided into three subfamilies, including B2_Mm2 (80,541 copies), B2_Mm1a (16,321 copies), and B2_Mm1t (35,812 copies). B2 SINE elements have been characterized to show a wide range of regulatory activities in mice, including acting as promoters (*Ferrigno et al., 2001*), insulator elements bound by CTCF (*Ichiyanagi et al., 2021*; *Lunyak et al., 2007*; *Schmidt et al., 2012*), or regulatory non-coding RNAs (*Hernandez et al., 2020*; *Karijolich et al., 2017*; *Karijolich et al., 2015*; *Schaller et al., 2020*; *Yakovchuk et al., 2009*). As the potential for B2_Mm2 SINEs to act as inducible enhancers has not yet been investigated, we decided to further investigate B2 SINEs in this context.

The B2_Mm2 subfamily showed strong evidence of enrichment within regions bound by STAT1, providing 2,122 total binding sites (odds ratio = 4.85). These B2_Mm2 elements show significantly higher localization near ISGs (p-value = $5.03 \times 10^{-52}$, odds ratio = 9.13) than interferon-repressed genes (IRGs) or nonresponsive genes, compared to unbound B2 elements or random genomic regions (*Figure 1B*, *Figure 1—figure supplement 1A–B*). We did not observe consistent enrichment of the B2_Mm1a and B2_Mm1t subfamilies over STAT1-bound regions (*Supplementary file 4*). Additionally, STAT1-bound B2_Mm2 elements are transcriptionally upregulated at the family level in response to IFNG stimulation (*Figure 1C*, *Figure 1—figure supplement 2*, *Supplementary file 5*). Although unbound B2_Mm2 elements are also transcriptionally active, we did not observe a significant increase in expression in response to IFNG stimulation. Taken together, these data indicate that thousands of B2_Mm2 elements show epigenetic and transcriptional evidence of IFNG-inducible regulatory activity in primary murine bone marrow derived macrophages.

We investigated the sequence features of each B2 SINE subfamily to determine the basis of IFNG-inducible activity. Given that B2 SINE elements have previously been associated with CTCF binding due to the presence of a CTCF motif harbored by most copies (*Schmidt et al., 2012*), we subdivided elements from each family based on occupancy by STAT1, CTCF, both factors, or neither factor based on ChIP-seq. Across each of these subsets, we looked for the presence of GAS or CTCF motifs (*Figure 2A*, *Figure 2—figure supplement 1*). As expected, all B2 subfamilies showed extensive ChIP-seq binding evidence of CTCF and the RAD21 cohesin subunit, coinciding with a CTCF motif (*Figure 2—figure supplement 1*). In contrast, only a subset of elements from the B2_Mm2 subfamily showed inducible binding of STAT1 (*Figure 2A*). Consistent with ChIP-seq evidence, STAT1-bound

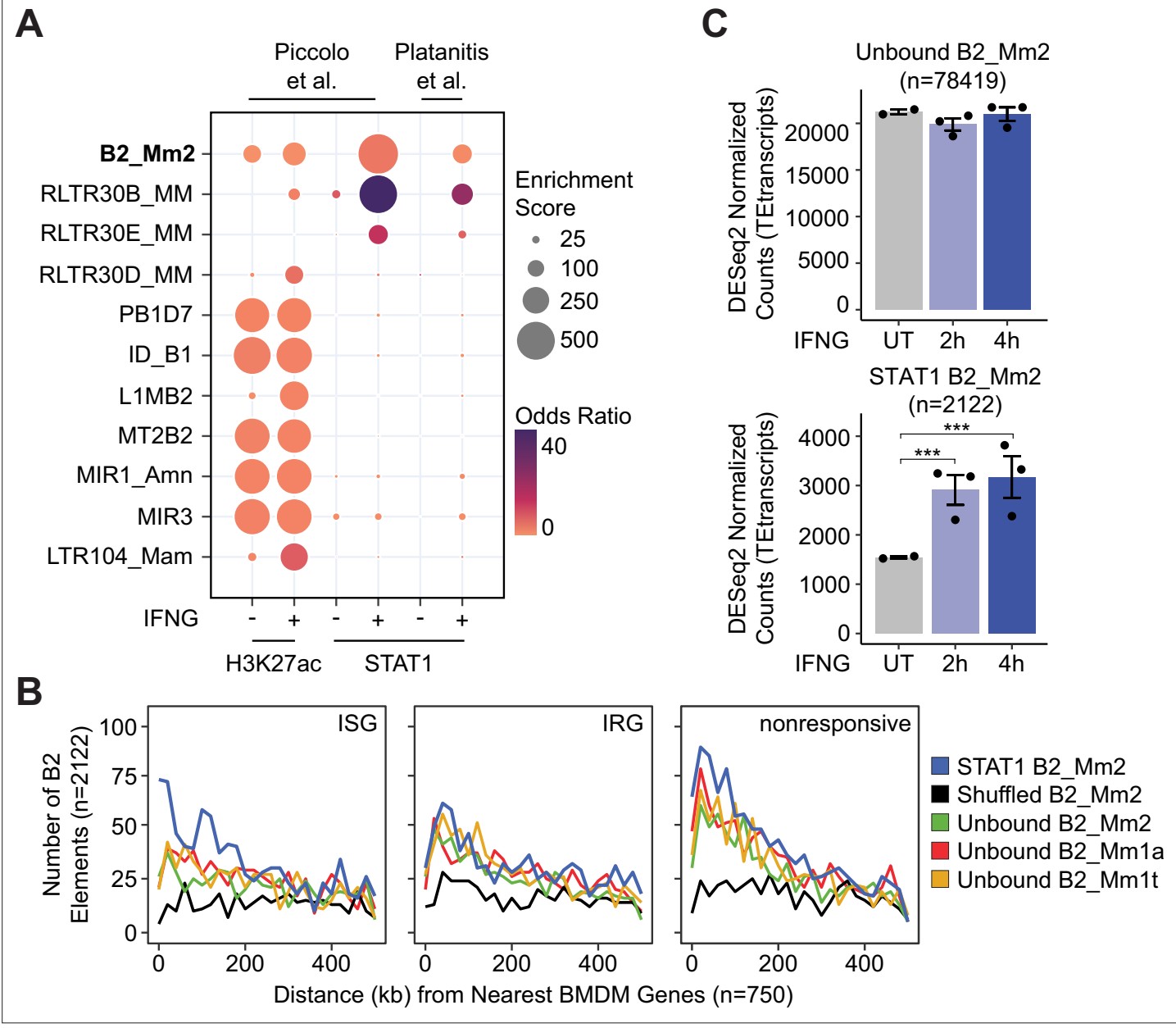

**Figure 1.** Identification of transposon-derived enhancers in innate immunity. (**A**) Bubble plot showing family-level enrichment of transposons within ChIP-seq peak regions. TE families enriched for STAT1 and H3K27ac ChIP-seq peaks are sorted by descending median Kimura distance. GIGGLE enrichment score is a composite of the product of both -log$_{10}$(p-value) and log$_{2}$(odds ratio). (**B**) Frequency histogram of absolute distances from STAT1-bound B2_Mm2 (blue, n=2,122), randomly shuffled B2_Mm2 (black, n=2,122), and randomly subset unbound B2_Mm2 (green, n=2,122), B2_Mm1a (red, n=2,122), and B2_Mm1t (yellow, n=2,122) elements to the nearest ISG (n=750), IRG (n=750), or nonresponsive gene (n=750). Data shown for (*Piccolo et al., 2017*) comparing expression in BMDMs stimulated with IFNG for 4 hr relative to untreated. (**C**) DESeq2 normalized counts showing immune-stimulated expression of unbound B2_Mm2 (top, n=78,419) and STAT1-bound B2_Mm2 (bottom, n=2122) elements in murine BMDMs. Data shown for untreated (n=2) BMDMs and BMDMs stimulated with IFNG for 2 hr (n=3) or 4 hr (n=3). Treatments are indicated by color. ***DESeq2 FDR adjusted p-value <0.0001. Error bars designate SEM. Data shown for (*Piccolo et al., 2017*). ISG: Interferon-stimulated gene; IRG: Interferon-repressed gene. BMDMs: Bone marrow derived macrophages. SEM: Standard error of mean.

The online version of this article includes the following figure supplement(s) for figure 1:

**Figure supplement 1.** Distances from each B2 element to the nearest ISG.

**Figure supplement 2.** IFNG-inducible TE expression in murine BMDMs.

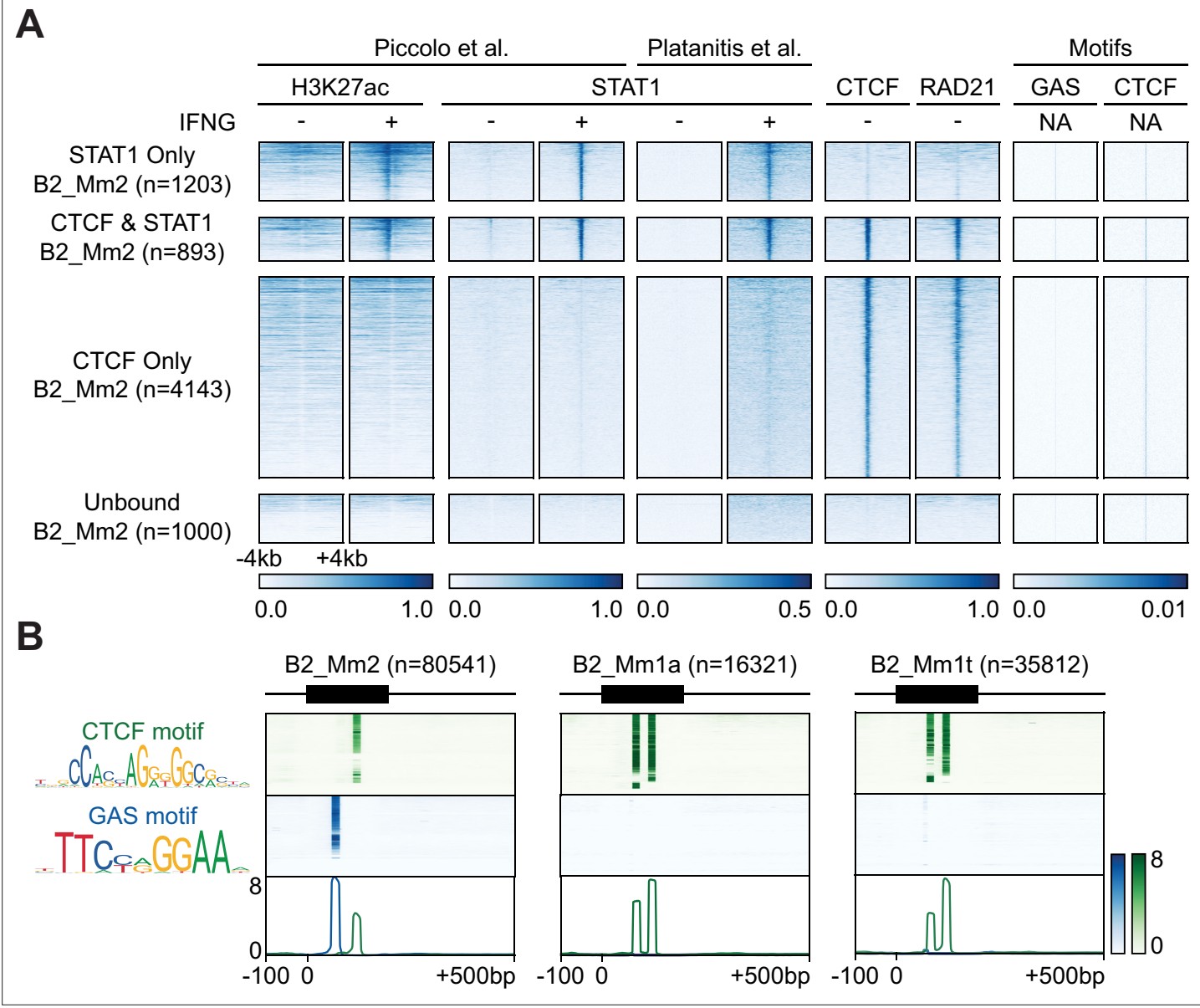

**Figure 2.** Epigenomic profiling of B2_Mm2. (**A**) Heatmaps showing CPM-normalized ChIP-seq signal and motif signal centered over B2_Mm2 elements bound only by STAT1 (n=1,203); B2_Mm2 bound by both STAT1 and CTCF (n=893); B2_Mm2 bound only by CTCF (n=4,143); and a random subset of unbound B2_Mm2 (n=1,000). Regions are sorted by descending mean CPM signal. Signal intensity is indicated below. CTCF track derived from *Gualdrini et al., 2022*. RAD21 track derived from *Cuartero et al., 2018*. (**B**) Schematic of GAS (blue) and CTCF (green) motifs present within extant B2_Mm2 (left, n=80,541), B2_Mm1a (middle, n=16,321), and B2_Mm1t (right, n=35,812) sequences. Heatmap intensity corresponds to motif matches based on the log likelihood ratio. Heatmaps are sorted by descending mean signal. Position weight matrices were obtained from JASPAR (*Fornes et al., 2020*). CPM: Counts per million. GAS: Gamma-IFN activated sequence.

The online version of this article includes the following figure supplement(s) for figure 2:

**Figure supplement 1.** STAT1 and CTCF occupancy overlapping all B2 elements in murine BMDMs.

**Figure supplement 2.** Multiple sequence alignment of B2 consensus sequences and B2_Mm2.Dicer1.

**Figure supplement 3.** Distribution of p-values for GAS motifs overlapping B2 elements.

B2_Mm2 elements contain both GAS and CTCF motifs, while B2_Mm1a/t elements only harbor CTCF motifs (*Figure 2B*, *Figure 2—figure supplement 2*). Within B2_Mm2, elements that are bound by STAT1 are significantly enriched for GAS motifs when compared against unbound elements (E-value $2.08 \times 10^{-71}$, *Supplementary file 6*). In addition, STAT1-bound B2_Mm2 elements contain stronger

sequence matches to GAS motifs compared to unbound B2_Mm2 elements and B2_Mm1a/t elements (*Figure 2A*, *Figure 2—figure supplement 3*). While B1_Mm1a elements show a partial match to the GAS motif, they show no evidence of STAT1 binding (*Figure 2B*, *Figure 2—figure supplement 2*). Therefore, elements of the B2_Mm2 subfamily are uniquely characterized by containing strong matches to GAS motifs that are associated with STAT1 binding activity.

Notably, we found that STAT1-bound B2_Mm2 elements are the only B2 elements that show an inducible H3K27ac signal associated with enhancer activity. In contrast, B2_Mm2 elements bound only by CTCF or unbound elements show minimal H3K27ac signal (*Figure 2A*). B2_Mm1a/t elements also show minimal STAT1 signal regardless of CTCF binding (*Figure 2—figure supplement 2*). This suggests that the binding of STAT1 to B2_Mm2 elements causes activation of enhancer activity including acetylation of H3K27. Thus, B2_Mm2 elements represent a distinct subclass of B2 SINE elements that exhibit IFNG-inducible enhancer activity.

## An intronic B2_Mm2 element functions as an inducible enhancer of Dicer1

Having established that B2_Mm2 elements are an abundant source of IFNG-inducible STAT1 binding sites in the mouse genome, we asked whether any of these elements have been co-opted to regulate expression of individual ISGs. We first assigned predicted enhancers to their predicted targets using the Activity by Contact (ABC) model (*Fulco et al., 2019*), which incorporates both epigenomic signal and Hi-C 3D interaction data to predict enhancer-gene targets. As input into the ABC model, we used publicly available ATAC-seq and Hi-C data from murine BMDMs stimulated with IFNG for 2 hours (*Platanitis et al., 2022*) and H3K27ac ChIP-seq and RNA-seq data (*Piccolo et al., 2017*). Focusing on a permissive set of 2,720 ISGs (FDR adjusted p-value <0.05, $log_2$FC >0), we identified 530 B2_Mm2 elements predicted to interact with 457 mouse ISGs (16.8% of the set of 2720 in this analysis; *Supplementary file 7*). Compared with a set of human ISGs from human CD14+ monocytes (*Qiao et al., 2016*), 393 of these 457 (86%) genes were only ISGs in mouse, and the remaining 64 (14%) were ISGs in both mouse and human.

The 393 mouse-specific ISGs predicted to be regulated by B2_Mm2 elements were significantly enriched for multiple immune related functions (GO:0002376, adjusted p-value = $6.023 \times 10^{-9}$; *Supplementary file 7*). We identified multiple examples of predicted B2_Mm2 target genes with established immune functions that showed mouse-specific IFNG-inducible expression (*Figure 3—figure supplement 1*, *Figure 3—figure supplement 2*), including dicer 1 ribonuclease III (*Dicer1*) (*Poirier et al., 2021*; *Figure 3A*), SET domain containing 6, protein lysine methyltransferase (SETD6) (*Levy et al., 2011*), DOT1-like histone lysine methyltransferase (*Kealy et al., 2020*), fumarate hydratase 1 (Fh1) (*Zecchini et al., 2023*), heat shock protein family A (Hsp70) member 1B (*Hspa1b*) (*Jolesch et al., 2012*), and NFKB inhibitor delta (*Nfkbid*) (*Souza et al., 2021*).

From this set, we decided to focus on a specific B2_Mm2 element located on Chromosome 12 within the first intron of *Dicer1*, which is an endonuclease responsible for recognizing and cleaving foreign and double stranded RNA that has been linked to innate immunity (*Chiappinelli et al., 2012*; *Gurung et al., 2021*; *MacKay et al., 2014*; *Poirier et al., 2021*). While the human ortholog *DICER1* does not show IFNG-inducible expression in human primary macrophages (*Qiao et al., 2016*; *Figure 3—figure supplement 3*, *Supplementary file 1*), mouse *Dicer1* shows a significant 50% upregulation in response to IFNG in primary mouse BMDMs (*Figure 3A*). This indicates that *Dicer1* is a mouse-specific ISG and likely acquired IFNG-inducible expression in the mouse lineage, potentially due to the co-option of the B2_Mm2 element as a species-specific IFNG-inducible enhancer. The intronic B2_Mm2 element (B2_Mm2.Dicer1) shows biochemical hallmarks of enhancer activity including inducible STAT1 and H3K27ac signal as well as constitutive binding by CTCF and RAD21 (*Figure 3B*). The element provides the only prominent nearby STAT1 binding site and is not present in rat or other mammals (*Figure 3B*). Therefore, we hypothesized that the B2_Mm2.Dicer1 element was co-opted as an IFNG-inducible enhancer of mouse *Dicer1*.

To experimentally test the potential enhancer activity of B2_Mm2.Dicer1, we used the mouse J774A.1 macrophage-like cell line, a commonly used model of murine immunity (*Lam et al., 2009*; *Ralph and Nakoinz, 1975*). We first confirmed using RT-qPCR that *Dicer1* shows 30–40% upregulation after 4 hr of IFNG treatment (*Figure 3C*). Given the STAT1 binding site and motif present in B2_Mm2. Dicer1, we also tested other cytokines that act through STAT-family transcription factors. We found

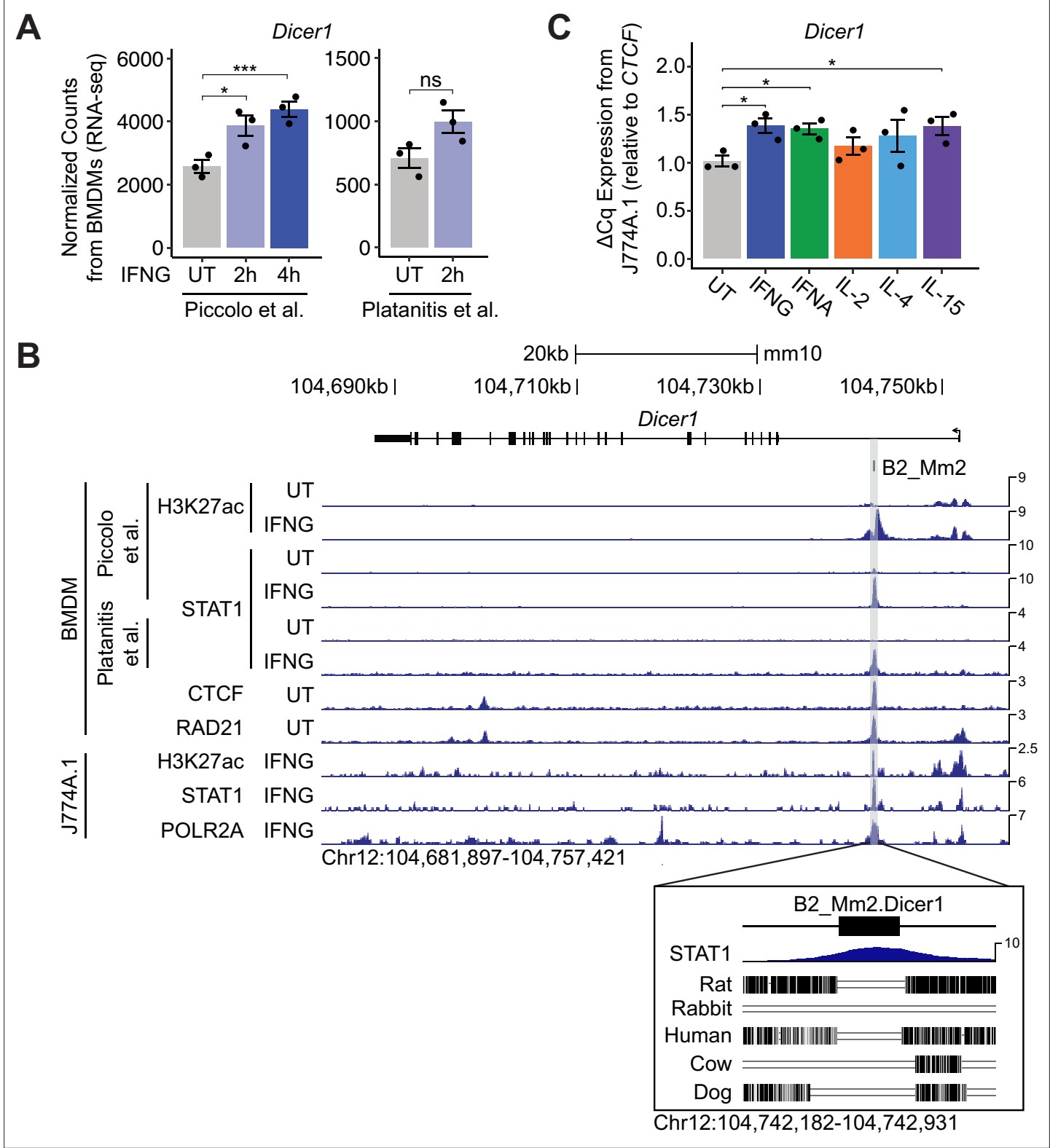

**Figure 3.** Identification of a putative B2_Mm2-derived enhancer for Dicer1. (**A**) DESeq2 normalized counts showing immune-stimulated expression of *Dicer1* in primary murine BMDMs. Data shown for untreated (n=3) BMDMs and BMDMs stimulated with IFNG for 2 hr (n=3) or 4 hr (n=3). ***DESeq2 FDR adjusted p-value <0.001, *DESeq2 FDR adjusted p-value <0.05. Error bars designate SEM. (**B**) Genome browser screenshot (http://genome.ucsc. edu) of the *Dicer1* locus (Chr12:104,681,897–104,757,421) showing CPM-normalized ChIP-seq tracks for primary murine BMDMs and immortalized macrophage line J774A.1. B2_Mm2.Dicer1 (Chr12:104,742,467–104,742,646) is highlighted in gray. Values on the right of each track correspond to

*Figure 3 continued on next page*

*Figure 3 continued*

signal maxima. Bottom inset shows B2_Mm2.Dicer1 with accompanying STAT1 (IFNG 2 h Piccolo et al.) ChIP-seq signal and conservation tracks for rat, rabbit, human, cow, and dog. CTCF track derived from **Gualdrini et al., 2022**. RAD21 track derived from **Cuartero et al., 2018**. (**C**) RT-qPCR of wild type untreated (gray, n=3) J774.A.1 cells and J774.A.1 cells stimulated with IFNG (blue, n=3), IFNA (green, n=3), IL-2 (orange, n=3), IL-4 (light blue, n=3), or IL-15 (purple, n=3) for 4 hr. Treatments are indicated by color. *p-value <0.05, Student's paired two-tailed *t*-test. BMDM: Bone-marrow-derived macrophage. SEM: Standard error of mean. CPM: Counts per million.

The online version of this article includes the following source data and figure supplement(s) for figure 3:

**Figure supplement 1.** Predicted B2_Mm2 enhancers associated with mouse-specific ISGs *Setd6, Dot1l, and Fh1*.

**Figure supplement 2.** Predicted B2_Mm2 enhancers associated with mouse-specific ISGs *Hspa1b* and *Nfkbid*.

**Figure supplement 3.** Constitutive expression of human *DICER1*.

**Figure supplement 4.** Validation of CRISPR B2_Mm2 knockout in J774.A1 cells.

**Figure supplement 4—source data 1.** Uncropped gel image from *Figure 3—figure supplement 4B*.

**Figure supplement 4—source data 2.** Uncropped gel image from *Figure 3—figure supplement 4D*.

**Figure supplement 4—source data 3.** Uncropped gel image from *Figure 3—figure supplement 4E*.

**Figure supplement 4—source data 4.** Uncropped gel image from *Figure 3—figure supplement 4F*.

**Figure supplement 4—source data 5.** Uncropped, labeled gel images from *Figure 3—figure supplement 4*.

**Figure supplement 5.** B2_Mm2 KO effect on the genomic landscape.

that 4 hr treatment of J774A.1 cells with IFNA, IL6, and IL4 all induced *Dicer1* expression to similar levels (30–40%), consistent with inducible regulation of *Dicer1* by JAK-STAT signaling, potentially through the STAT binding site present in B2_Mm2.Dicer1.

We next used CRISPR to generate clonal J774A.1 lines harboring homozygous deletions of the B2_Mm2.Dicer1 element. We delivered guide RNAs targeting the flanking boundaries of B2_Mm2.Dicer1 along with recombinant Cas9 by electroporation (**Figure 3—figure supplement 4A–C**), isolated clones by limiting dilution, and screened clonal lines for homozygous deletions by PCR (**Figure 3—figure supplement 4D–F**). We isolated two clonal cell lines with a homozygous knockout of B2_Mm2.Dicer1, along with multiple wild-type (WT) J774A.1 clonal lines that were not electroporated to control for potential effects of clonal expansion (**Figure 3—figure supplement 4F**). We used RT-qPCR to compare *Dicer1* expression levels and inducibility by IFNG in knockout and WT clones. WT clones showed consistent inducible expression, while both knockout clonal lines showed a complete lack of inducible expression (**Figure 4A**). These experiments demonstrate that B2_Mm2.Dicer1 acts as an IFNG-inducible enhancer of *Dicer1* in J774A.1 cells.

## B2_Mm2.Dicer1 impact on the genomic regulatory landscape

We used RNA-seq to study the genome-wide effects of the B2_Mm2.Dicer1 element in both knockout clones and three control wild-type clones which were also isolated by limiting dilution. Consistent with the RT-qPCR results, we found that *Dicer1* showed significant IFNG-inducible upregulation in all WT clones but that this induction was completely ablated in B2_Mm2.Dicer1 KO clones (**Figure 4B**). Notably, the RNA-seq normalized count data revealed that expression of *Dicer1* were also significantly reduced in untreated KO cells ($\log_2 FC = -0.43$, FDR-adjusted p-value = $9.246 \times 10^{-4}$) (**Figures 4B and 5A**, **Figure 3—figure supplement 5B–C**). Focusing on the IFNG-treated condition, *Dicer1* was significantly downregulated in KO cells compared to WT cells ($\log_2 FC = -0.80$, FDR-adjusted p-value = $3.393 \times 10^{-15}$). The deletion of the element did not affect inducibility of other ISGs in the 5 Mb locus including the nearby highly induced *SerpinA3* ISGs locus (**Figure 4C–D**). This indicates that the B2_Mm2.Dicer1 element specifically regulates both basal expression levels of *Dicer1* and inducible expression by IFNG. While the enhancer deletion on *Dicer1* expression levels had a modest downregulating effect, this effect was specific and consistent across individually edited clones, particularly under IFNG-stimulated conditions (**Figure 4C–D**, **Figure 3—figure supplement 5D–E**).

Genome-wide, there were 101 genes that showed greater significance than *Dicer1* when testing for differential expression between IFNG-treated KO and WT cells (out of 3,567 genes with FDR-adjusted p-value <0.05; **Supplementary file 8**). Out of these 101 genes, 24 showed higher variability than *Dicer1* (based on DESeq2 $\log_2 FC$ standard error) between individual clones (**Figure 3—figure supplement 5B–E**), consistent with intrinsic clonal transcriptional variation revealed by the limiting

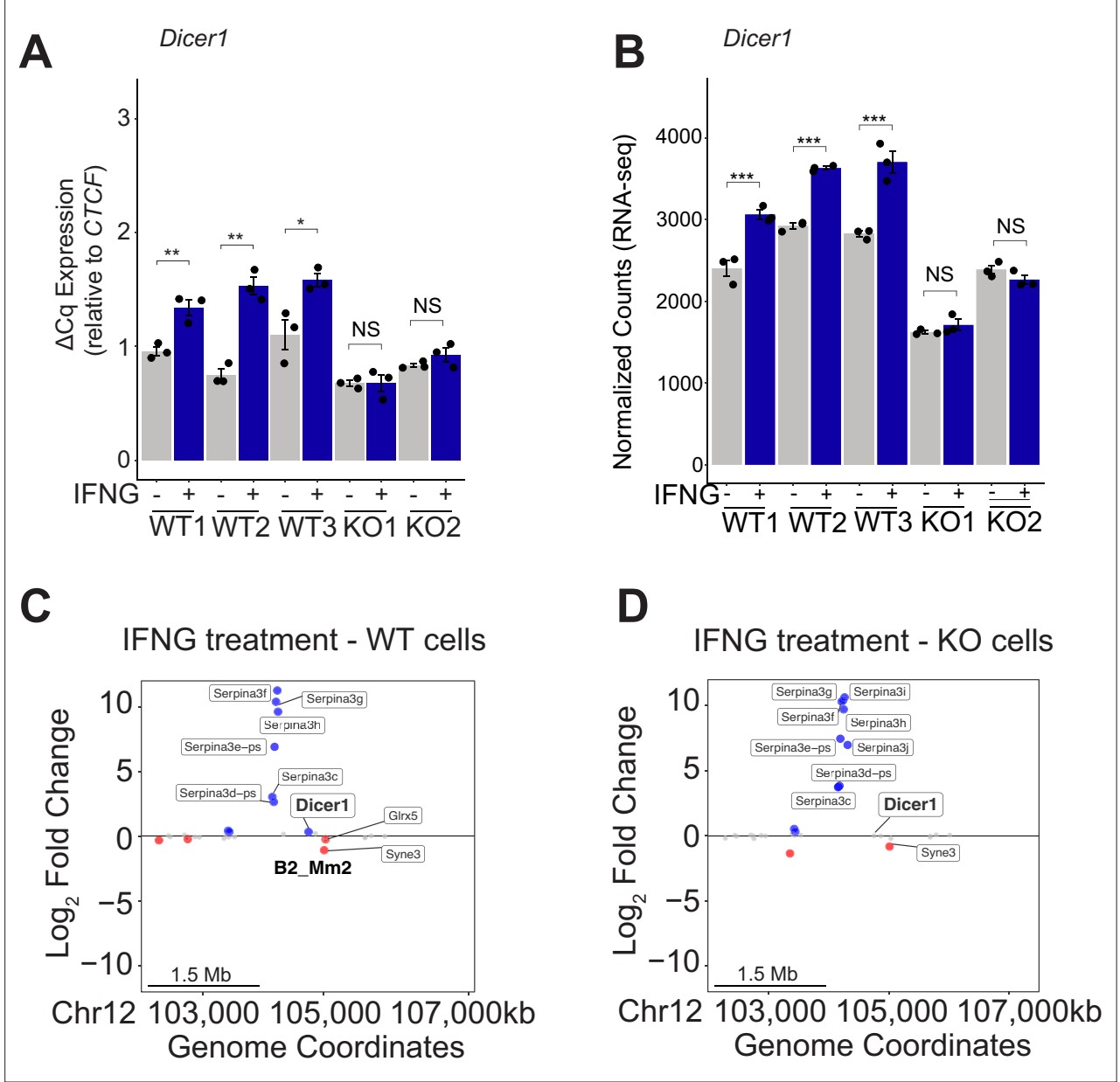

**Figure 4.** B2_Mm2 in the genomic landscape. (**A**) RT-qPCR of *Dicer1* expression in untreated and IFNG-treated cell lines across 3 clonal WT lines J774A.1 cells and 2 B2_Mm2.Dicer1 knockout (KO) lines, with three replicate treatments per cell line. *Dicer1* expression was normalized relative to *CTCF*. Treatments are indicated by color. *p-value <0.05, **p-value <0.01, ***p-value <0.001, Student's paired two-tailed *t*-test. (**B**) DESeq2 normalized counts of *Dicer1* expression in each clonal WT and B2_Mm2.Dicer1 KO J774A.1 cell line. Treatments are indicated by color. ***DESeq2 FDR adjusted p-value <0.001. (**C**) Distance plot visualizing changes in gene expression in wild type J774A.1 cells in response to IFNG over a 5 Mb window centered on B2_Mm2.Dicer1. Significantly downregulated (log₂FC <0, FDR adjusted p-value <0.05) genes are shown in red while significantly upregulated (log₂FC >0, FDR adjusted p-value <0.05) genes are shown in blue. *Dicer1* is labeled, as well as significantly IFNG-regulated genes within 1 Mb. (**D**) Same as in (**C**) but visualizing changes in gene expression in KO J774A.1 cells in response to IFNG.

dilution and/or CRISPR editing process (**Nahmad et al., 2022**; **Westermann et al., 2022**). Thirty-five of these genes showed upregulation in the KO cells, suggesting that they could be silencing targets of *Dicer1* that become upregulated upon *Dicer1* downregulation. However, given the relatively modest effect of *Dicer1* especially in the untreated condition, further experiments would be necessary to establish these genes as targets of *Dicer1*.

We confirmed the absence of intronic enhancer activity in B2_Mm2.Dicer1 knockout cells. We used CUT&TAG (**Kaya-Okur et al., 2020**) to profile H3K27ac, phosphorylated RNA polymerase II subunit

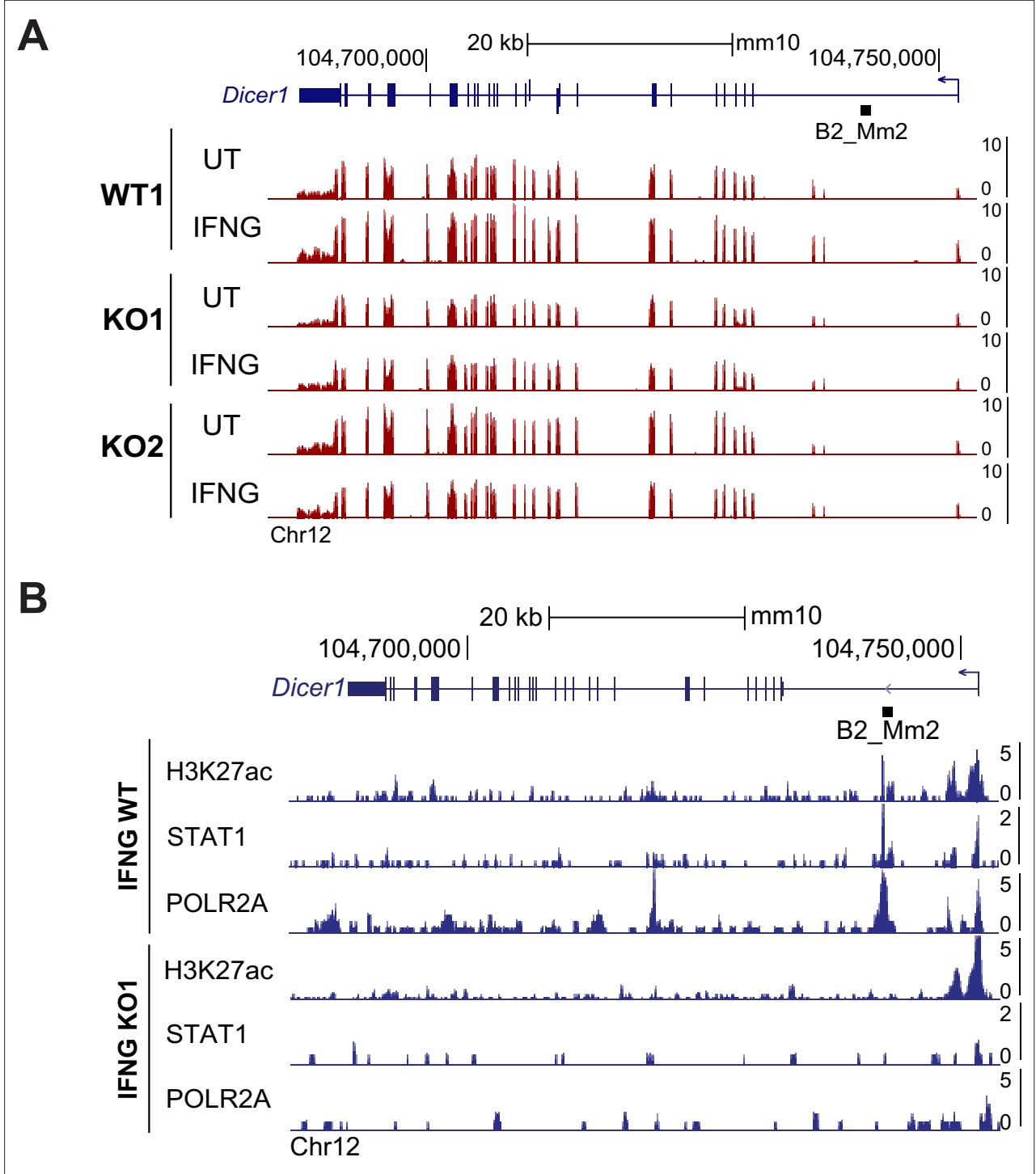

**Figure 5.** B2_Mm2 impacts local chromatin profile. (**A**) Genome browser screenshot of the *Dicer1* locus visualizing CPM-normalized expression in WT and B2_Mm2.Dicer1 KO J774A.1 cells. Values on the right of each track correspond to signal maxima. B2_Mm2.Dicer1 is represented as a black box (not drawn to scale). (**B**) Genome browser screenshot of the *Dicer1* locus showing CUT&TAG data from bulk WT and B2_Mm2.Dicer1 KO J774A.1 cells. CPM: Counts per million.

A (POLR2A), and STAT1. In IFNG-stimulated WT cells, the B2_Mm2.Dicer1 element shows prominent H3K27ac, STAT1, and POLR2A signal. However, these signals are completely lost in the knockout clone (*Figure 5B*). Collectively, these experiments confirm that B2_Mm2.Dicer1 has been co-opted to function as an IFNG-inducible enhancer that regulates *Dicer1*.

Given the B2_Mm2.Dicer1 element is bound by POLR2A in WT cells, we asked whether the B2_Mm2.Dicer1 element may alter transcription by affecting usage of different splice sites or polyadenylation sites, which would be consistent with pause site activity (*Jonkers and Lis, 2015*). We examined transcript isoform-level expression changes in KO cells in both untreated and treated conditions and found multiple transcripts that showed the same trend as the gene-level analysis, where most expressed transcripts are downregulated in both basal and IFNG-treated conditions and show lack of inducibility in KO cells (*Supplementary file 9*). These findings are consistent with the element acting primarily as an IFNG-inducible enhancer without any major effect on alternative splicing. However, further experiments such as CDK9 inhibition and profiling of nascent transcription in stimulated conditions (*Gressel et al., 2017*; *Laitem et al., 2015*) are necessary to establish whether the element affects transcriptional elongation.

Considering that the B2_Mm2 enhancer is specific to rodents, we examined the regulatory landscape of the human *DICER1* locus. Our analysis of RNA-seq data from human primary monocytes treated with IFNG for 24 hr (*Qiao et al., 2016*) indicated that human *DICER1* expression is not induced by IFNG (*Supplementary file 1*, *Figure 3—figure supplement 3*). However, ChIP-seq data from IFN-treated monocytes from the same group (*Qiao et al., 2013*) showed multiple inducible STAT1 binding sites within the human *DICER1* locus, including one originating from a primate-specific TE (LTR27) (*Figure 3—figure supplement 3*). Although these binding sites do not correlate with inducible *DICER1* expression in the matched RNA-seq dataset, they suggest human *DICER1* may be inducible under different conditions. An analysis of an independent dataset generated from another donor (*McCann et al., 2022*) supported the inducible expression of *DICER1* (log$_2$FC = 0.91 and FDR adjusted *P*-value = 3.0 × 10$^{-4}$; *Supplementary file 1*). Thus, while the evidence for inducible human *DICER1* expression is inconsistent, our analyses indicate that human *DICER1* has independently evolved primate-specific binding STAT1 binding sites, which may also confer inducible regulation.

## Discussion

B2 SINE elements are abundant in the mouse genome and they have been widely studied due to their substantial influence on genome regulation and evolution. B2 elements have contributed non-coding RNAs inducible by stress or infection (*Allen et al., 2004*; *Karijolich et al., 2017*; *Karijolich et al., 2015*; *Li et al., 1999*; *Schaller et al., 2020*; *Walters et al., 2009*; *Wick et al., 2003*; *Williams et al., 2004*), splicing signals (*Kress et al., 1984*), promoter elements (*Ferrigno et al., 2001*), and CTCF-bound insulator elements (*Ichiyanagi et al., 2021*; *Lunyak et al., 2007*; *Schmidt et al., 2012*). Our study reveals a new subclass of B2 elements that have IFNG-inducible enhancer activity. These elements, which belong to the B2_Mm2 subfamily, contain strong binding sites for both STAT1 and CTCF, are marked by H3K27ac, and have the potential to exert inducible enhancer activity on nearby genes.

Given the abundance of B2 elements and their potential to cause pathological regulatory rewiring, many B2 elements are targeted for SETDB1/H3K9me3-mediated epigenetic repression which inhibits their regulatory potential (*Gualdrini et al., 2022*). Therefore, the functional impact of B2 elements on the mouse epigenome remains unclear. By using CRISPR to generate knockout cells of a B2_Mm2 element, we demonstrated that B2 elements can be co-opted to act as inducible enhancer elements in the context of IFNG stimulation. While our experiments were conducted in the J774.A1 immortalized cell line, we confirmed that thousands of B2_Mm2 elements including B2_Mm2.Dicer1 show strong transcriptional and epigenetic signatures of inducible enhancer activity in multiple primary macrophage epigenomic datasets.

We identified hundreds of B2_Mm2-derived enhancers that are predicted to regulate genes displaying IFNG-inducible expression in mouse cells but not human cells, which supports their role in facilitating lineage-specific evolution of the IFNG-inducible regulatory network. However, we also identified a subset of target genes that show inducible expression in both species, suggesting independent evolution or turnover of regulatory elements that could serve similar regulatory functions as the B2_Mm2 enhancers. For instance, we identified an intronic STAT1 binding site derived from a primate-specific TE in the human *DICER1* locus. While we did not uncover consistent evidence supporting IFNG-inducible regulation of human *DICER1*, these observations align with the concept of convergent regulatory evolution, in which similar expression patterns are perpetuated by the co-option of lineage-specific TEs (*Choudhary et al., 2020*; *Sundaram et al., 2014*). Therefore, as B2_Mm2

elements shaped the evolution of rodent immune regulatory networks, individual co-option events may have mediated either divergence or preservation of gene expression patterns.

Our identification of a B2_Mm2 element as an intronic enhancer of *Dicer1* potentially uncovers a novel regulatory feedback loop that controls *Dicer1* function related to TE silencing. Previous studies have demonstrated that *Dicer1* cleaves double-stranded RNAs including those derived from B2 SINE transcripts (*Fan et al., 2021*). In *Dicer1* knockout embryonic stem cells, TE-derived transcripts are upregulated (*Bodak et al., 2017*), and upregulation of B2-derived double-stranded RNAs causes activation of the IFN response (*Gurung et al., 2021*). Therefore, *Dicer1* is important for defense against aberrant TE upregulation. We speculate that co-option of the B2_Mm2 element as an enhancer of *Dicer1* facilitates upregulation of *Dicer1* in response to conditions that drive TE upregulation, such as infection or stress.

In human, dysregulation of *DICER1* is associated with a wide range of pathologies ranging from DICER1 syndrome, cancer, neurological diseases such as Parkinson's disease, and autoimmune disorders such as rheumatoid arthritis (*Theotoki et al., 2020*). While *DICER1* has a highly conserved function as an endonuclease involved in RNA interference, our work highlights that orthologs of *DICER1* have undergone lineage- or species-specific regulatory evolution that may drive underappreciated differences in function across species. While we found that both human and mouse orthologs of *DICER1* have STAT1 binding sites, the underlying cis-regulatory architecture is not conserved and likely results in different expression patterns. This could have significant implications when developing and testing RNA-based therapeutics in mouse genetic models, which may elicit distinct *Dicer1*-mediated responses due to species-specific enhancers such as B2_Mm2.Dicer1.

In conclusion, our work adds to a growing body of evidence highlighting the co-option of lineage-specific TEs for the regulation of ISGs. Previous genomic and experimental studies in human (*Chuong et al., 2016*) and cow cells (*Kelly et al., 2022*) have revealed independent co-option of TEs as IFNG-inducible enhancer elements. Interestingly, while endogenous retroviruses have been described as a prevalent source of TE-derived inducible enhancers in human, their contribution is relatively minor in cows and mice. Instead, we found that the B2_Mm2 SINE subfamily is the predominant source of TE-derived inducible enhancers in mouse, and the Bov-A2 SINE subfamily is the predominant source in cow (*Kelly et al., 2022*). These findings indicate that the acquisition of STAT1-associated GAS motifs and enhancer activity is not limited to a specific type of TE. In human cells, Alu SINE sequences are transcriptionally activated upon infection (*Jang and Latchman, 1989*; *Panning and Smiley, 1993*), although their epigenetic impact remains unexplored. It remains unclear whether the infection-inducible regulatory activity promotes SINE replication, and it is possible that the emergence of these motifs is coincidental and does not affect TE fitness. Nevertheless, our work supports the idea that TEs have been repeatedly co-opted as IFNG-inducible enhancers throughout mammalian evolution, contributing to the rewiring of immune regulatory networks.

## Materials and methods

### Sequences

A list of all primer sequences and gRNA sequences can be found in *Supplementary file 10*.

### RNA-seq re-analysis

RNA-seq data (single-end reads) from primary murine BMDMs stimulated with 100 ng/mL IFNG for 2 or 4 hr (*Piccolo et al., 2017*) or 10 ng/mL IFNG for 2 hr (*Platanitis et al., 2019*) were downloaded from SRA using fasterq-dump v2.10.5 (*NCBI, 2022*). Adapters and low-quality reads were trimmed using BBDuk v38.05 (*Bushnell, 2018*) using options '*ktrim = r k=34 mink = 11 hdist = 1 qtrim = r trimq = 10 tpe tbo*'. Library quality was assessed using FastQC v0.11.8 (*Andrews, 2018*) and MultiQC v1.7 (*Ewels et al., 2016*), and trimmed reads were aligned to the mm10 assembly using HISAT2 v2.1.0 (*Kim et al., 2019*) with option '`--no-softclip`'. Only uniquely aligned fragments (MAPQ ≥ 10) were retained using samtools v1.10 (*Li et al., 2009*). Aligned fragments were assigned to the complete mm10 Gencode vM18 (*Frankish et al., 2021*) annotation in an unstranded manner using featureCounts v1.6.2 (*Liao et al., 2014*) with options '*-p -O -s 0 t exon -g gene_id*', and differentially expressed genes between IFNG-stimulated and unstimulated cells were called using DESeq2 v1.26.0 (*Love et al., 2014*). For most analyses, ISGs and IRGs were defined as genes with a false discovery

rate (FDR) adjusted p-value of at least 0.05 and log$_2$FC greater than 0 and less than zero, respectively. Nonresponsive genes were defined using the following cutoffs: baseMean greater than 100; FDR adjusted p-value greater than 0.90; and absolute log$_2$FC less than 0.10. Interferon stimulation was confirmed by gene ontology analysis using gProfiler (last updated 05/18/2022) with FDR adjusted p-value <0.05 (*Raudvere et al., 2019*). We additionally aligned RNA-seq data from human CD14$^+$ monocytes stimulated with 100 U/mL IFNG for 24 hr to the hg38 assembly and identified differentially expressed genes using Gencode v38 (*Frankish et al., 2021*) with the methods described above.

## ChIP-seq re-analysis

ChIP-seq data (single-end reads) from primary murine BMDMs (*Cuartero et al., 2018*; *Gualdrini et al., 2022*; *Piccolo et al., 2017*; *Platanitis et al., 2019*) were downloaded from SRA using fasterq-dump v2.10.5 (*NCBI, 2022*). Adapters and low-quality reads were trimmed using BBDuk v38.05 (*Bushnell, 2018*) using options '*ktrim = r k=34 mink = 11 hdist = 1 qtrim = r trimq = 10 tpe tbo*'. Library quality was assessed using FastQC v0.11.8 (*Andrews, 2018*) and MultiQC v1.7 (*Ewels et al., 2016*), and trimmed reads were aligned to the mm10 assembly using BWA-MEM v0.7.15 (*Li, 2013*). Low quality and unmapped reads were filtered using samtools v1.10 (*Li et al., 2009*), and duplicates were removed with Picard MarkDuplicates v2.6.0 (*Broad Institute, 2016*). Peak calling was performed with MACS2 v2.1.1 (*Liu, 2014*) using options '*--gsize mm –pvalue 0.01 –bdg –SPMR –call-summits*'. bigWigs corresponding to read pileup per million reads for visualization on the UCSC Genome Browser (*Kent et al., 2002*). Where possible, only peaks overlapping more than one replicate were retained for further analysis. To confirm whether STAT1 peaks were enriched for their associated binding motifs, we ran XSTREME v5.4.1 (*Grant and Bailey, 2021*) using options '*--minw 6 --maxw 20 –streme-nmotifs 20 –align center*' querying against the JASPAR CORE 2018 vertebrates database (*Fornes et al., 2020*).

## Transposable element analysis

To identify TE families enriched for STAT1 and H3K27ac peaks, we used GIGGLE v0.6.3 (*Layer et al., 2018*) to create a database of all TE families annotated in the mm10 genome according to Dfam v2.0 (*Storer et al., 2021*) annotation. STAT1 and H3K27ac ChIP-seq peaks were then queried against each TE family in the database. GIGGLE applies the Fisher's exact test to assess family-level enrichment, attributing an odds ratio, Fisher's two tailed p-value, and a GIGGLE combo score combining the two values. We only retained TE families that met the following criteria: (1) number of total annotated elements >100; (2) number of elements overlapping a ChIP-seq peak >30; (3) odds ratio >3; and (4) a GIGGLE combo score >100. Results were visualized as a bubble plot where the filtered TE families were sorted by ascending Kimura divergence according to RepeatMasker (*Smit et al., 2019*) output. Reported odds ratios and p-values are derived from Fisher's exact test. For further analysis, we intersected STAT1 peaks with the full TE annotation or B2 elements specifically using BEDTools v2.28.0 (*Quinlan and Hall, 2010*). For the heatmap visualizations using deepTools v3.0.1 (*Ramírez et al., 2016*), signal from counts per million- (CPM) normalized bigWigs was plotted over a subset of B2_Mm2 elements that are bound only by STAT1, CTCF, both, or neither. We additionally visualized ChIP-seq signal over all B2_Mm2, B2_Mm1a, and B2_Mm1t elements by descending average signal, excluding elements with zero overlapping signal.

To assess whether STAT1-bound B2_Mm2 elements are enriched near ISGs, we sorted all ISGs and IRGs by descending and ascending log$_2$FC, respectively, and retained the top 750 genes. We additionally randomly subset for 750 nonresponsive genes. The absolute distance to the nearest ISG, IRG, or nonresponsive gene was determined for all STAT1-bound B2_Mm2 elements using BEDTools v2.28.0 (*Quinlan and Hall, 2010*). Randomly shuffled STAT1-bound B2_Mm2 as well as randomly subset, unbound B2_Mm2, B2_Mm1a, and B2_Mm1t were included as controls. Statistical significance was determined for the first 20 kb bin using Fisher's exact test with BEDTools v2.28.0 (*Quinlan and Hall, 2010*).

To identify TE families that are differentially expressed in response to IFNG in primary murine BMDMs, we realigned the RNA-seq data to the mm10 reference genome using HISAT2 v2.1.0 (*Kim et al., 2019*) with options '*-k100 –no-softclip*'. Aligned reads were assigned to TE families using TEtranscripts v2.1.4 (*Jin et al., 2015*) with options '*--sortByPos –mode multi –iteration 100 –stranded no*'. TEtranscripts allows for quantification of TE expression at the family level and does not discriminate between individual elements within a family. To differentiate unbound and STAT1-bound B2_Mm2

elements, we generated a custom TE annotation file compatible with TEtranscripts that includes all TEs annotated in Dfam v2.0 (*Storer et al., 2021*) but annotates STAT1-bound B2_Mm2 elements as an independent subfamily. Differentially expressed TE families between IFNG-stimulated and unstimulated cells were identified using DESeq2 v1.26.0 (*Love et al., 2014*). TE families with an FDR less than 0.05 and an absolute $\log_2$FC greater than 0.50 were considered as differentially expressed. These relaxed thresholds were used to better enable the identification of differentially expressed TE families where only a subset of elements are inducibly expressed and the majority are only lowly expressed.

We identified putative STAT1 and CTCF binding sites genome-wide using FIMO v5.0.3 (*Grant et al., 2011*) with a *p*-value cutoff of $1\times10^{-4}$ (heatmaps) or 1 (B2 box-and-whisker). For all motif analyses, binding motif position-weight matrices for STAT1 and CTCF were obtained from the JASPAR CORE 2018 vertebrate database (*Fornes et al., 2020*). To visualize motif presence over all B2 elements, repeat 5' start coordinates were recalculated based on their alignment to the consensus according to RepeatMasker annotations. Motif presence was visualized as a heatmap using deepTools v3.0.1 (*Ramírez et al., 2016*), and elements were sorted by descending average signal. We additionally aligned the consensus sequences for B2_Mm2, B2_Mm1a, and B2_Mm1t from Repbase v24.02 and the sequence for B2_Mm2.Dicer1 using MUSCLE v3.8.1551 (*Edgar, 2004*). Predicted STAT1 and CTCF motifs were identified using FIMO v.5.0.3 (*Grant et al., 2011*), and base changes relative to the canonical binding motifs were highlighted according to the weight of each individual base in the position-weight matrices. Finally, we filtered for STAT1-bound B2_Mm2 elements that were non-overlapping, non-nested, and unique and ran AME v5.4.1 (*McLeay and Bailey, 2010*) using a subset of unbound B2_Mm2 elements as the background control with options '--*kmer 2 –method fisher –scoring avg*'.

## ATAC-seq re-analysis

Paired-end ATAC-seq data from primary murine BMDMs stimulated with 10 ng/mL IFNG for 2 hr (*Platanitis et al., 2022*) were downloaded from SRA using fasterq-dump v2.10.5 (*NCBI, 2022*). Adapters and low-quality reads were trimmed using BBDuk v38.05 (*Bushnell, 2018*) with options '*ktrim = r k=34 mink = 11 hdist = 1 tpe tbo qtrim = r trimq = 10*'. Library quality was assessed using FastQC v0.11.8 (*Andrews, 2018*) and MultiQC v1.7 (*Ewels et al., 2016*), and trimmed r eads were aligned to the mm10 assembly using Bowtie 2 v2.2.9 (*Langmead and Salzberg, 2012*) with options '--*end-to-end --very-sensitive -X 1000 --fr*', and only uniquely mapping reads with a minimum MAPQ of 10 were retained. Fragments aligning to the mitochondrial genome were removed, and duplicates were removed using Picard MarkDuplicates v2.6.0 (*Broad Institute, 2016*). Aligned fragments were shifted +4/–5 using deepTools alignmentSieve v3.0.1 with option '--*ATACshift*' (*Ramírez et al., 2016*) and used to call ATAC-seq peaks with an FDR <0.05 using MACS2 v2.1.1 (*Liu, 2014*) with options '--*SPMR -B --keep-dup all --format BAMPE –call-summits*'.

## Hi-C re-analysis

Paired-end Hi-C data from primary murine BMDMs stimulated with 10 ng/mL IFNG for 2 hr (*Platanitis et al., 2022*) were downloaded from SRA using fasterq-dump v2.10.5 (*NCBI, 2022*). Library quality was assessed using FastQC v0.11.8 (*Andrews, 2018*) and MultiQC v1.7 (*Ewels et al., 2016*), and reads were aligned to the mm10 assembly using BWA-MEM 0.7.17 (*Li, 2013*) with arguments '-*SP*', and the resulting bam file was converted to pairsam format pairtools parse v0.2.2 (https://github.com/mirnylab/pairtools; *Goloborodko, 2019*). Technical replicates were merged using pairtools merge v0.2.2. Duplicate reads were marked using pairtools dedup v0.2.2, and only aligned fragments with pairtools classification 'UU' or 'UC' were retained using pairtools filter v0.2.2 resulting in approximately 600 M pairs. A Knight Ruiz (KR)-normalized, Arima restriction site-aware Hi-C matrix was prepared using juicer pre v1.22.01 (*Dudchenko et al., 2017*) at 5 kb resolution and fitted to a powerlaw distribution in preparation for running the Activity-by-Contact (ABC) model (*Fulco et al., 2019*).

## Gene-enhancer target prediction by Activity-by-Contact (ABC) analysis

We applied the Hi-C matrix (*Platanitis et al., 2022*) in conjunction with ATAC-seq (*Platanitis et al., 2022*) and H3K27ac (*Piccolo et al., 2017*) data from IFNG-stimulated BMDMs to predict enhancer activity using the Activity-by-Contact (ABC) model (*Fulco et al., 2019*). The ABC model predicts enhancer-gene contacts by leveraging epigenomic and chromatin interaction capture data. Each

potential enhancer-gene contact is assigned an ABC interaction score that depends on the activity of the enhancer by ATAC-seq and H3K27ac ChIP-seq in addition to the likelihood of contact by Hi-C. We ran the ABC model as previously described (*Fulco et al., 2019*). In brief, we first identified candidate enhancer elements using makeCandidateRegions.py with options *'--peakExtendFromSummit 250 – nStrongestPeaks 150000'* and quantified activity using run.neighborhoods.py. Predicted enhancer-gene pairs were attributed an ABC interaction score using predict.py with options *'--hic_type juicebox –hic_resolution 5000 –scale_hic_using_powerlaw –threshold 0.02 –make_all_putative'*. Only enhancer regions with an ABC interaction score over 0.001 were considered for subsequent analysis.

## Orthology analysis

We used BioMart with human Ensembl v105 annotation (*Cunningham et al., 2022*) to identify high confidence, one-to-one orthologs in mouse for each identified human ISG ($\log_2$FC >0, FDR adjusted p-value <0.05). A union set consisting of mouse and human-to-mouse ISGs was generated, and each ISG was identified as mouse-specific, human-specific, or shared according to induction status. To determine how broadly B2_Mm2 has shaped murine innate immune responses, we identified 344 unique B2_Mm2 elements fully overlapping enhancers predicted to interact with 706 ISGs using the ABC model with a minimum ABC score of 0.001. The proportion of ISGs predicted to interact with a putative B2_Mm2 enhancer were plotted according to species status. We independently identified 926 ISGs with at least one of 655 STAT1-bound B2_Mm2 elements (irrespective of ABC) within 50 kb of the transcriptional start site.

## Cell line passing and interferon treatments

J774A.1 mouse cells were purchased from ATCC and were cultured in DMEM supplemented with 1 X penicillin-streptomycin and 10% fetal bovine serum. J774A.1 cells were routinely passaged using 0.25% Trypsin-EDTA and cultured at 37 °C and 5% CO2. All IFNG treatments were performed using 100 ng/mL recombinant mouse IFNG (R&D Systems #485-MI-100). Cells were confirmed to be Mycoplasma-free by the Barbara Davis Center for Childhood Diabetes BioResources Core Molecular Biology Unit at the University of Colorado Anschutz Medical Center. The identity of the cells was verified using the ATCC Cell Line Authentication Service (STR profiling).

## Cytokine panel

All treatments were carried out using a 4 hr time period. IL-4 (Sigma-Aldrich #I1020-5UG) was added to a final concentration of 1 ng/mL, recombinant IFNA (R&D Systems #12100–1) to a final concentration of 1000 U/mL, recombinant IL-2 (R&D Systems #402 ML-020) to a final concentration of 100 ng/mL, and recombinant IL-6 (Sigma-Aldrich #I9646-5UG) to a final concentration of 20 ng/mL in accordance with manufacturers' recommendations. RNA was extracted using an Omega Mag-Bind Total RNA Kit (Omega Bio-Tek #M6731-00) and analyzed via RT-qPCR.

## Design of gRNA constructs

Two gRNA sequences were designed to flank each side of B2_Mm2.Dicer1 in order to delete the element and generate knockout J774A.1 cells via SpCas9 (Integrated DNA Technologies #1081060). All gRNA sequences were also verified to uniquely target the locus of interest using the UCSC BLAT tool (*Kent, 2002*) against the mm10 genome assembly.

## Generation of CRISPR KO cell lines

After gRNAs were designed, we used Alt-R Neon electroporation (1400 V pulse voltage, 10 ms pulse width, 3 pulses total) with four different combinations of the gRNAs to target B2_Mm2. One set of guides was found to produce the expected doubles-stranded cuts on both sides of the element in the bulk electroporated cell populating using gel electrophoresis. Clonal lines were isolated using the array dilution method and screened for the expected homozygous deletion using primers flanking the B2_Mm2.Dicer1 element. Clonal lines homozygous for the deletion were further validated using one flanking primer and one primer internal to B2_Mm2.Dicer1. To determine deletion breakpoint sequences, PCR products flanking each deletion site were cloned into a sequencing vector using the CloneJET PCR Cloning Kit (Thermo Fisher Scientific #K1231) and transformed into 5-alpha Competent *E. coli* (New England Biolabs #C2987H). Plasmid DNA was harvested using the EZNA

Omega Plasmid DNA Mini Kit I (Omega Bio-Tek #D6942-02), and the sequence of each construct was verified by Sanger sequencing (Quintara Biosciences, Fort Collins, CO; Genewiz, South Plainfield, NJ). Sequencing results were visualized by aligning to the mm10 reference genome using BLAT (*Kent, 2002*). We identified two clonal lines homozygous for the B2_Mm2.Dicer1 deletion for further experimentation.

## Quantifying *Dicer1* expression using RT-qPCR

Real-time quantitative polymerase chain reaction (RT-qPCR) was used to quantify *Dicer1* expression in WT and knockout cell lines. WT J774A.1 cells used in RT-qPCR are all biological replicates that underwent the same single cell seeding process as the KOs to serve as a control. Forward and reverse primers were designed for *CTCF*, *Dicer1*, and *Gbp2b*. Each set of primers was designed using a combination of tools from NCBI Primer BLAST (*Ye et al., 2012*), Benchling (Benchling), and IDT RT-qPCR Primer Design. The final primer sequences chosen were confirmed to uniquely bind to the desired target sequence using BLAT (*Kent, 2002*). RT-qPCR reactions were prepared using the Luna Universal One-Step RT-qPCR Kit (New England Biolabs #E3005S) according to the manufacturer's instructions. RT-qPCR data were analyzed using *CTCF* as a housekeeping gene. A Cq, deltaCq, and deltaDeltaCq value were obtained for each well. These values were averaged to arrive at a mean deltaDeltaCq expression value for each treatment and genotype condition. Standard deviation and a two-tailed Student's *t*-test were then calculated for each treatment and genotype condition. A p-value of less than 0.05 demonstrates there is a statistically significant difference in gene expression levels between two treatment and/or genotype conditions.

## J774A.1 RNA-seq library preparation

The Zymo Quick RNA Miniprep Plus Kit (Zymo Research #R1504) was used to extract RNA from J774A.1 cells for all treatments except for the cytokine panel which used the Omega RNA Extraction Kit (Omega Bio-Tek #M6731-00). WT J774A.1 cells used in RNA-seq and all downstream analysis were clones that underwent single-cell expansions of from wild-type J774A.1 cells. All RNA lysates and single-use aliquots of extracted RNA were stored at –80 °C until library preparation. RNA integrity was quantified with High Sensitivity RNA TapeStation 4200 (Agilent). Libraries were generated using the KAPA mRNA HyperPrep Kit (KAPA Biosystems #08098123702) according to the manufacturer's protocol. The final libraries were pooled and sequenced on a NovaSeq 6000 as 150 bp paired-end reads (University of Colorado Genomics Core).

## J774A.1 RNA-seq analysis

Adapters and low quality reads were first trimmed using BBDuk v38.05 (*Bushnell, 2018*). Library quality was assessed using FastQC v0.11.8 (*Andrews, 2018*) and MultiQC v1.7 (*Ewels et al., 2016*) and trimmed reads were aligned to the mm10 assembly using HISAT2 v2.1.0 (*Kim et al., 2019*) with option '--rna-strandness RF'. Only uniquely aligned fragments were retained, and technical replicates were merged using samtools v1.10 (*Li et al., 2009*). CPM-normalized, stranded bigWigs were generated using deepTools bamCoverage v3.0.1 (*Ramírez et al., 2016*) and visualized using the UCSC Genome Browser (*Kent et al., 2002*). Aligned fragments were assigned to the mm10 refseq gene annotation in a reversely stranded manner using featureCounts v1.6.2 (*Liao et al., 2014*) with options '-t exon -s 2', and differentially expressed genes were called using DESeq2 v1.26.0 (*Ramírez et al., 2016*). We analyzed every individual pairwise comparison to determine the effects due to both treatment and genotype. Untreated and wild type conditions were defined as the reference level. Log$_2$FC values were shrunken using the apeglm function v1.8.0 (*Zhu et al., 2019*) for visualization across Chromosome 12 as a distance plot.

To determine whether the B2_Mm2.Dicer1 element acts as a regulator of splicing, we conducted transcriptome-guided transcript assembly on all WT and B2_Mm2.Dicer1 KO J774 RNA-seq alignments individually using Stringtie v1.3.3b with options '--rf -j 5'. Individual GTF files were merged into a single file with option '--merge'. Reads were subsequently aligned against the merged transcriptome using Salmon v1.9.0 with options '--validateMappings –rangeFactorizationBins4 –gcBias'. Transcript-level quantification data for each WT and KO sample were used for differential expression analysis with DESeq2 v1.26.0 (*Ramírez et al., 2016*).

## CUT&Tag

CUT&Tag datasets were generated using a protocol from *Kaya-Okur and Henikoff, 2020*; *Kaya-Okur et al., 2020* with an input of 100–500 k cells and the following modifications: pAG-Tn5 (EpiCypher #15–1017) was diluted 1:40 in nuclease-free water containing 20 mM HEPES pH 7.5, 300 mM NaCl, 0.5 mM spermidine, 0.01% digitonin, and a protease inhibitor tablet, and libraries were amplified for 14 cycles. The following primary antibodies were used: rabbit IgG (1:1000, EpiCypher #13–0042), rabbit anti-H3K27ac (1:100), rabbit anti-pRPB1-Ser5 (1:100, Cell Signaling Technology #13523 S), rabbit anti-STAT1 (1:100, Cohesion Biosciences #CPA3322), rabbit anti-pSTAT1-Ser727 (1:100, Active Motif #39634), rabbit anti-CTCF (1:100, EMD Millipore #07-729-25UL). Guinea pig anti-rabbit IgG (1:100, Antibodies-Online #ABIN101961) was used as a secondary antibody. CUTANA pAG-Tn5 (EpiCypher #15–1017) was added to each sample following primary and secondary antibody incubation. Pulldown success was measured by Qubit dsDNA High Sensitivity (Invitrogen) and TapeStation 4200 HSD5000 (Agilent) before proceeding to library preparation. Pulldowns were concentrated and pooled using KAPA Pure Beads (Roche). The final pooled libraries were quantities with TapeStation 4200 HSD5000 and sequences on an Illumina NovaSeq 6000 as 150 bp paired-end reads (University of Colorado Genomics Core).

## CUT&Tag analysis

Adapters and low quality reads were first trimmed using BBDuk v38.05 (*Bushnell, 2018*). Library quality was assessed using FastQC v0.11.8 (*Andrews, 2018*) and MultiQC v1.7 (*Ewels et al., 2016*). Trimmed reads were then aligned to the mm10 assembly using BWA-MEM v0.7.15 (*Li, 2013*) and samtools (*Li et al., 2009*) retained only uniquely aligned fragments (MAPQ $\geq$ 10). Peaks were called without a control file using MACS2 v2.1.1 (*Liu, 2014*). bigWigs corresponding to read pileup per million reads for visualization on the UCSC Genome Browser (*Kent et al., 2002*).

## External datasets

Publicly available data were downloaded from public repositories using fasterq-dump from the NCBI SRA Toolkit. RNA-seq datasets were obtained from GSE84517, GSE115434, GSE84691, GSE176562, and GSE43036. ChIP-seq datasets were obtained from GSE84518, GSE108805, GSE189971, and GSE115433. Hi-C and ATAC-seq datasets were obtained from SRA using accession PRJNA694816.

## Data access

Raw and processed sequencing data generated in this study have been submitted to the NCBI Gene Expression Omnibus (GEO) with accession number GSE202574.

## Code Availability

UCSC Genome browser sessions and all code available at https://genome.ucsc.edu/s/coke6162/B2_SINE_enhancers_Horton_et_al and https://github.com/coke6162/B2_SINE_enhancers (copy archived at *Kelly, 2023*).

# Acknowledgements

We thank the University of Colorado Genomics Shared Resource and BioFrontiers Computing core for technical support during this study. Funding EBC was supported by the National Institutes of Health (1R35GM128822), the Alfred P Sloan Foundation, the David and Lucile Packard Foundation, and the Boettcher foundation. IH was supported by the Boettcher foundation.

## Additional information

### Funding

| Funder | Grant reference number | Author |
| --- | --- | --- |
| National Institutes of Health | 1R35GM128822 | Edward B Chuong |

| Funder | Grant reference number | Author |
|---|---|---|
| David and Lucile Packard Foundation | | Edward B Chuong |
| Boettcher Foundation | | Isabella Horton<br>Edward B Chuong |
| Alfred P. Sloan Foundation | | Edward B Chuong |

The funders had no role in study design, data collection and interpretation, or the decision to submit the work for publication.

## Author contributions

Isabella Horton, Conceptualization, Formal analysis, Investigation, Methodology, Writing - original draft; Conor J Kelly, Conceptualization, Investigation, Visualization, Methodology, Writing - original draft; Adam Dziulko, Methodology; David M Simpson, Investigation, Methodology, Project administration; Edward B Chuong, Conceptualization, Supervision, Funding acquisition, Project administration, Writing - review and editing

## Author ORCIDs

Conor J Kelly http://orcid.org/0000-0003-2410-7892
Edward B Chuong http://orcid.org/0000-0002-5392-937X

## Decision letter and Author response

Decision letter https://doi.org/10.7554/eLife.82617.sa1
Author response https://doi.org/10.7554/eLife.82617.sa2

# Additional files

## Supplementary files

• Supplementary file 1. DESeq2& GO results for murine BMDMs genes stimulated with IFNG for 2 or 4 hours.

• Supplementary file 2. Murine BMDM ChIP-seq motif enrichment using XSTREME.

• Supplementary file 3. Coordinates for STAT1-bound TEs in murine BMDMs.

• Supplementary file 4. Family level TE enrichment over murine BMDM ChIP-seq using GIGGLE.

• Supplementary file 5. DESeq2 results for TE expression in murine BMDMs stimulated with IFNG for 2 or 4 hours.

• Supplementary file 6. Differential B2_Mm2 motif enrichment analysis using AME.

• Supplementary file 7. Putative B2_Mm2 enhancers identified using the Activity-by-Contact model or by proximity.

• Supplementary file 8. DESeq2 gene-level results for IFNG-stimulated WT and B2_Mm2.Dicer1 KO J774A.1 cells.

• Supplementary file 9. DESeq2 transcript-level results for IFNG-stimulated WT and B2_Mm2.Dicer1 KO J774A.1 cells.

• Supplementary file 10. Primer and gRNA sequences used in this study.

• MDAR checklist

## Data availability

Raw and processed sequencing data generated in this study have been submitted to the NCBI Gene Expression Omnibus (GEO) with accession number GSE202574.

The following dataset was generated:

| Author(s) | Year | Dataset title | Dataset URL | Database and Identifier |
|---|---|---|---|---|
| Horton I, Kelly CJ, Simpson DM, Chuong EB | 2022 | Mouse SINE B2 elements function as IFN-inducible enhancer elements | https://www.ncbi.nlm.nih.gov/geo/query/acc.cgi?acc=GSE202574 | NCBI Gene Expression Omnibus, GSE202574 |

The following previously published datasets were used:

| Author(s) | Year | Dataset title | Dataset URL | Database and Identifier |
|---|---|---|---|---|
| Piccolo V, Curina A, Genua M, Ghisletti S, Simonatto M, Sabò A, Amati B, Ostuni R, Natoli G | 2017 | Opposing macrophage-polarization programs show extensive epigenomic and transcriptional cross-talk [RNA-Seq] | https://www.ncbi.nlm.nih.gov/geo/query/acc.cgi?acc=GSE84517 | NCBI Gene Expression Omnibus, GSE84517 |
| Platanitis E, Decker T | 2019 | Transcription profile analysis of wild type and Irf9-/- bone marrow derived macrophages in response to type I and type II interferons | https://www.ncbi.nlm.nih.gov/geo/query/acc.cgi?acc=GSE115434 | NCBI Gene Expression Omnibus, GSE115434 |
| Qiao Y, Giannopoulou E, Zhang T | 2016 | RNAseq to profile IFNg response in human primary monocytes | https://www.ncbi.nlm.nih.gov/geo/query/acc.cgi?acc=GSE84691 | NCBI Gene Expression Omnibus, GSE84691 |
| McCann KJ, Christensen S, McGuire PJ, Myles IA, Zerbe CS, Li P, Sukumar G, Dalgard CL, Leonard WJ, McCormick BA, Holland SM | 2021 | IFNg Regulates NAD+ Metabolism in Human Monocytes | https://www.ncbi.nlm.nih.gov/geo/query/acc.cgi?acc=GSE176562 | NCBI Gene Expression Omnibus, GSE176562 |
| Qiao Y, Li Y, Giannopoulou E | 2013 | Synergistic Activation of Inflammatory Cytokine Genes by Priming of Regulatory DNA Elements for Increased Transcription in Response to TLR Signaling | https://www.ncbi.nlm.nih.gov/geo/query/acc.cgi?acc=GSE43036 | NCBI Gene Expression Omnibus, GSE43036 |
| Piccolo V, Curina A, Genua M, Ghisletti S, Simonatto M, Sabò A, Amati B, Ostuni R, Natoli G | 2017 | Opposing macrophage-polarization programs show extensive epigenomic and transcriptional cross-talk [ChIP_narrow] | https://www.ncbi.nlm.nih.gov/geo/query/acc.cgi?acc=GSE84518 | NCBI Gene Expression Omnibus, GSE84518 |
| Merkenschlager M, Dharmalingam G, Cuartero S | 2013 | Transcriptional control of macrophage inducible gene expression by cohesin [ChIP-Seq II] | https://www.ncbi.nlm.nih.gov/geo/query/acc.cgi?acc=GSE108805 | NCBI Gene Expression Omnibus, GSE108805 |
| Gualdrini F, Polletti S, Simonatto M, Prosperini E, Natoli G | 2022 | H3K9 trimethylation in active compartments optimizes stimulus-regulated transcription by restricting usage of CTCF sites in SINE-B2 repeats [ChIP-Seq] | https://www.ncbi.nlm.nih.gov/geo/query/acc.cgi?acc=GSE189971 | NCBI Gene Expression Omnibus, GSE189971 |
| Platanitis E, Decker T | 2019 | STAT1, STAT2 and IRF9 transcription factor binding analysis in wild type and Irf9-/- bone marrow derived macrophages in response to type I and type II interferons | https://www.ncbi.nlm.nih.gov/geo/query/acc.cgi?acc=GSE115433 | NCBI Gene Expression Omnibus, GSE115433 |
| Max Perutz Labs | 2021 | 3D chromatin rearrangements in response to Interferon treatment | https://www.ncbi.nlm.nih.gov/sra/PRJNA694816 | NCBI Sequence Read Archive, PRJNA694816 |

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
