## [Editor Report]

This important paper will be of interest to scientists studying evolutionary divergence of immune responses and those studying how transposable elements rewire transcriptional regulatory networks. Using a combination of computational and experimental approaches, this work describes a new class of rodent-specific transposons that can act as enhancers of immune genes in mice.

---

## [Decision Letter]

**Decision letter after peer review:**

Thank you for submitting your article "Mouse B2 SINE elements function as IFN-inducible enhancers" for consideration by *eLife*. Your article has been reviewed by 3 peer reviewers, and the evaluation has been overseen by a Reviewing Editor and Molly Przeworski as the Senior Editor. The following individual involved in review of your submission has agreed to reveal their identity: Kenji Ichiyanagi (Reviewer #3).

There was consensus among the reviewers that the topic is of broad interest and the data were rigorously generated. The reviewers, however, also felt that some of the claims require additional support. A request for new analyses of the current dataset in hand could potentially blunt these concerns. Depending on the results of these additional analyses, a single new experiment would be required to support the inference that the focal B2 element acts as a Dicer1 enhancer. Please see below for specific revisions requested.

Essential revisions:

1) It is not clear whether the Dicer1 locus is unique. Are there other loci that have both a nearby B2_Mm2 element and a binary difference between inducibility in mouse versus human cells? It would be helpful to understand whether there other B2_Mm2 insertions that could contribute to a mouse-specific IFGN-sensitivity. Adding DEseq values from human RNAseq data the authors already use (current references 10 and/or 37) for identifiable human orthologs to Table S7 would thus strengthen their conclusions. If there are many potential candidates, the authors should discuss the rationale for selecting Dicer1 in particular.

2) The results of Serpina3g and Serpina3F gene expression in the authors' knockout cells are very interesting. However, the authors focus almost exclusively on Serpina3g and Serpina3F, which makes it difficult to understand what is happening genome wide. Are other IFNΓ-induced genes (including those not on chromosome 12) similarly affected at the level of basal or induced transcription? How many genes are different in WT versus KO cells, both at basal and induced states? Does this correlate with their CUT&TAG data shown in Figure 5? By focusing only on nearby genes (Serpina3g and Serpina3F), the authors hint that this may be a long-range regulatory effect, "potentially mediated by the CTCF binding activity of the element" that they removed. But by only focusing on two nearby IFNΓ-induced genes, their data do not rule out the (also potentially quite interesting) possibility that there may be a more indirect role for this TE site or Dicer1 in basal transcription of IFNΓ-induced genes or IFNΓ-mediated gene expression.

3) The specificity of the KO effect on the Dicer1-Serpina region of chromosome 12 is not clear. Without analysis of the complete RNA-seq and CUT&RUN datasets, it is difficult to rule out a more global effect (i.e., beyond chromosome 12). If these new analyses yield evidence of specificity of the KO lines, the reviewers will be satisfied. If not, the reviewers request an additional manipulation: KO of an intron element of equivalent size to the original deletion, KO of a different B2 element – apparently there is one 2kb away, or even better, replacement of the B2 element with one that lacks cGAS motifs (though the final suggestion is likely too technically challenging). Determining whether there are changes in the basal or induced levels of Dicer1/Serpina genes in this additional line would serve as an important control for the KO experiment. Providing more data on other genes throughout the genome in WT and KO cells, which the authors have generated but do not include in the manuscript, would help distinguish between these models.

4) There are high levels of POLR2A occupancy at the B2_Mm2.Dicer1 element in induced WT cells. Could this be a Pol2 pause site? Could deletion of this element lead to a change in Pol2 occupancy and change Dicer1 expression independent of enhancer activity? To probe such questions, the reviewers requested that the authors directly test the possibility that the intronic B2 element actually acts as a regulator of splicing or transcriptional elongation. Careful analysis of the Dicer1-mapping reads from the RNA-seq data – or RT-qPCR – could resolve this concern.

5) Figure 4F – The authors claim that "deletion of B2_Mm2.Dicer1 also has a significant repressive effect on the IFNΓ-inducible expression of Serpina genes." However, the basal levels of Serpina3f/Serpina3g are significantly reduced upon this deletion compared to WT. Furthermore, expression of Serpina genes in the KO cell lines significantly increase upon IFNΓ stimulation, suggesting that they still show inducible expression despite the B2_Mm2.Dicer1 deletion. The authors should compare the magnitude of induction before and after stimulation between the WT and KO cell lines to determine if there is indeed a repressive effect on inducible expression of Serpina genes.

[Editors' note: further revisions were suggested prior to acceptance, as described below.]

Thank you for resubmitting your work entitled "Mouse B2 SINE elements function as IFN-inducible enhancers" for further consideration by *eLife*. Your revised article has been evaluated by Molly Przeworski (Senior Editor) and a Reviewing Editor.

The reviewers appreciated your many additional experiments/analyses as well as the edits to the text. The manuscript has been improved substantially but there are some remaining issues that need to be addressed.

1. All three reviewers felt that the response to Critique#1 was insufficient. There are several places in the manuscript referring to the significance of the Mouse B2 SINE elements contributing to lineage-specific immune response. For example, the introduction highlights that "a key open question is whether the co-option of TEs as immune regulatory elements is evolutionarily widespread as a mechanism driving divergence of innate immune responses."

The reviewers (or at least two of them) did not expect to see additional KO experiments but did expect the authors to highlight a handful of other examples where STAT1 binding to a proximal B2 SINE element is found in a mouse-specific ISG only. If no other such examples are available to highlight, then it's currently difficult to discern how generalizable this relationship is between rodent-specific B2 SINE elements and mouse-specific ISGs.

Furthermore, the authors highlight a limitation of the current human cell dataset – specifically the limitation of sampling gene expression only 24 hours after interferon treatment (new paragraph starting line 310) in human cells. However, in line 232, the same dataset is used to support the earlier statement that "the human ortholog DICER1 does not show IFNΓ-inducible expression in human primary macrophages (Qiao et al. 2016)…mouse DICER1 shows a significant 50% upregulation in response to IFNΓ in primary mouse BMDMs…."

This earlier statement is substantially weakened by the final Results paragraph. This issue needs to be resolved to support the major claim of the paper that a lineage-specific TE is responsible for a lineage-specific immune response, echoed in the abstract: "B2 elements…exemplifies how lineage-specific TEs can facilitate evolutionary divergence of innate immune regulatory networks."

2. In the response to reviewer comments under critique#2, it was stated that "…the other differentially expressed genes in the KOs most likely represent off-target/stochastic changes, that are commonly seen across separate clonal isolations." And as communicated in the manuscript: Line 288: "The other dysregulated genes showed no discernable physical or functional pattern and also showed high variability between individual clones…consistent with intrinsic clonal transcriptional variation…" Whether this is true for all 101 genes should be clarified. It seems surprising given that at least some genes should be indirect targets of Dicer1. Finally, does Dicer1 show less variability than these other genes? If so, that should be stated.

3. Finally, one reviewer was concerned that the intronic location of B2 SINE means that the authors cannot delineate whether the element acts as an enhancer or instead a regulator of transcriptional elongation.

---

## [Author Response]

Essential revisions:1) It is not clear whether the Dicer1 locus is unique. Are there other loci that have both a nearby B2_Mm2 element and a binary difference between inducibility in mouse versus human cells? It would be helpful to understand whether there other B2_Mm2 insertions that could contribute to a mouse-specific IFGN-sensitivity. Adding DEseq values from human RNAseq data the authors already use (current references 10 and/or 37) for identifiable human orthologs to Table S7 would thus strengthen their conclusions. If there are many potential candidates, the authors should discuss the rationale for selecting Dicer1 in particular.

We agree with the reviewers that it would be useful to investigate whether other B2 elements contribute to mouse-specific IFN-inducible expression, which may indicate whether the *Dicer1* example is unique or part of a larger species-specific network of genes regulated by B2 elements. As suggested, we have re-analyzed a publicly available human RNA-seq dataset to identify human ISGs in CD14^+^ monocytes stimulated with IFNΓ for 24 hours. A more detailed description of our analysis is provided below. In summary, we classified mouse ISGs based on whether those genes were also ISGs based on the human RNA-seq data, and integrated mouse Hi-C data to predict enhancer/gene interactions. We found that B2_Mm2 elements are associated nearby hundreds of mouse-specific ISGs, consistent with a role in regulating a network of mouse ISGs. However, we note several caveats below, including the observation that B2_Mm2 elements are also associated with many non mouse-specific ISGs, and cannot be causally tied to gene regulation without additional experimental perturbation.

Using RNA-seq data of IFNΓ-stimulated human CD14+ monocytes (GEO accession GSE84691) (Qiao et al., 2016), we identified a total of 2314 human ISGs (FDR adjusted *p*-value < 0.05, log_2_ fold change > 0), 1453 of which had a 1:1 ortholog assignment to a mouse gene based on ENSEMBL Biomart (Cunningham et al., 2022). We integrated the mouse and human datasets to classify mouse genes into 3 categories:

i) "mouse-specific ISGs" (i.e., genes that have a 1:1 mouse/human ortholog but are only ISGs in mouse OR mouse ISGs with no direct ortholog in human)

ii) "shared ISGs" (i.e., genes that have a 1:1 mouse/human ortholog and are ISGs in both species)

iii) "human-specific ISGs" (i.e., genes that have a 1:1 mouse/human ortholog but are only ISGs in human).

The DESeq2 values for this dataset are now reported in Supplementary file 1. Notably human *DICER1* does not appear to be induced (Supplementary file 1, Figure 3 —figure supplement 1). To compare this set to mouse ISGs, we used ISGs defined from BMDMs stimulated with IFNΓ for 4 hours from the Piccolo et al. dataset (GSE84517) analyzed in our previous submission (Piccolo et al., 2017). From this dataset, we defined 2720 mouse genes as ISGs (FDR adjusted *p*-value < 0.05, log_2_FC > 0). We compared this set of genes to the 1453 human ISGs defined above (i.e., genes that are an ISG in human CD14+ monocytes and have a direct 1:1 human:mouse orthology relationship). Out of the 2720 mouse ISGs, 2266 genes (83.3%) were only ISGs in mouse (including 1467 genes with a 1:1 human ortholog and 799 genes with no 1:1 ortholog), and 454 (16.6%) were ISGs in both mouse and human. Conversely, out of the 2314 human ISGs, 1860 genes (80.4%) were only ISGs in human (including 999 genes with a 1:1 mouse ortholog and 861 genes with no 1:1 ortholog). The relatively low fraction of ISGs shared across species is consistent with a previous study comparing ISGs induced by type I interferon α across vertebrates (Shaw et al., 2017), where a core of 62 ISGs were conserved across species (in our IFNΓ datasets, 56 of these are induced in either mouse or human, 40 of which are induced in both species).

We next sought to determine how broadly B2_Mm2 has shaped murine innate immune responses using our list of mouse-specific, human-specific, and shared ISGs. We first assigned predicted enhancers to their predicted targets using the Activity-by-Contact (ABC) model (Fulco et al., 2019), which incorporates both epigenomic signal and Hi-C 3D interaction data to predict enhancer-gene targets. As input into the ABC model, we used publicly available ATAC-seq and Hi-C data (PRJNA694816) (Platanitis et al., 2022) from murine BMDMs stimulated with IFNΓ for 2 hours along with the H3K27ac ChIP-seq and RNA-seq datasets included in the manuscript to identify putative enhancer-gene contacts. We identified 50264 putative enhancer-ISG contacts attributed a minimum ABC likelihood score of 0.01, with an average of 14 enhancers for each gene

represented in our list of mouse- and human-to-mouse ISGs. We subsequently identified 393 "mouse-specific" 142 "human-specific", and 64 "shared" ISGs associated with at least one B2_Mm2 enhancer. A list of all mouse and human-to-mouse ISGs along with the top predicted B2_Mm2-derived enhancer can be found in Supplementary file 7.

Of 2266 mouse-specific ISGs, 393 (17.3%) were predicted to be regulated by at least one B2_Mm2 element. We confirmed that the B2_Mm2.Dicer1 element was predicted by ABC to target *Dicer1* (a mouse-specific ISG) with an ABC likelihood score of 0.127508 (Supplementary file 7). This finding is consistent with the idea that B2_Mm2 elements have facilitated the evolution of the mouse-specific ISG regulatory network. However, we also found that B2_Mm2 elements are predicted to regulate 64 out of 454 (14.1%) genes that are shared ISGs in both mouse and human, and 142 out of 999 (14.2%) of genes that are only ISGs in human (Author response image 1). Additionally, we did not observe a difference in the distribution of ABC scores between mouse- and human-specific ISGs for B2_Mm2-derived enhancers as well as nonresponsive TE families (Author response image 2). This indicates that the association with a B2_Mm2-derived enhancer is not strongly predictive that a gene is a mouse-specific ISG.

There are a few notable caveats with this analysis. (1) the human RNA-seq dataset was generated by a different lab and derived from cells stimulated with IFNΓ for 24 hours, whereas the two mouse RNA-seq datasets collected cells 2 or 4 hours after IFNΓ stimulation. Previous studies have shown that transcriptional responses to type I and II IFNs vary across timepoints (Piccolo et al., 2017; Sumida et al., 2022). Genes that are not induced in human cells after 24 hours may potentially be induced at 4 hours. As such, the number of shared ISGs is likely an underestimate. Careful analysis of human and mouse datasets derived from cells collected at similar timepoints may improve identification of species-specific ISGs and potentially result in a larger difference in the number of B2_Mm2 elements predicted to drive species-specific ISG expression. (2), we used a relaxed ABC likelihood score threshold of 0.01 such that the number of B2_Mm2-derived enhancer-gene contacts identified likely represent the maximum possible using these data. These predictions also did not distinguish between elements that show constitutive or IFN-inducible regulation, so it is possible that some B2 elements may have act as constitutive rather than IFN-inducible enhancers. (3) we were able to confidently identify only 1453 (62.8%) human-to-mouse, one-to-one ISG orthologs. Careful analysis of all potential orthologs and paralogs (such as species-specific gene family expansions) may better reveal the full extent to which B2_Mm2 has contributed to species-specific immune regulatory evolution.

In summary, while our reanalysis is indeed consistent with a slightly increased enrichment of B2_Mm2 elements near mouse-specific ISG (e.g., 17.3% of mouse-specific ISGs are predicted to be regulated by B2_Mm2 elements compared to 14.1% of shared human/mouse ISGs), there are many caveats that we feel are outside the scope of this study to address and therefore we have opted not to include these analyses in the study.

We initially decided to study the B2_Mm2 within the *Dicer1* intron for several reasons that were outlined in the main text, including the species-specific IFN-inducible expression of *Dicer1* and strong epigenomic signature of inducible enhancer activity (STAT1, H3K27ac, CTCF) of the specific B2_Mm2 element. We have clarified our process for selecting the element as below, incorporating the revised ABC analysis to assign enhancers to genes through 3D interactions as described above.

“Having established that B2_Mm2 elements are an abundant source of IFNΓ-inducible STAT1 binding sites in the mouse genome, we asked whether any of these elements have been co-opted to regulate expression of individual ISGs. We first assigned predicted enhancers to their predicted targets using the Activity by Contact (ABC) model (Fulco et al., 2019), which incorporates both epigenomic signal and Hi-C 3D interaction data to predict enhancer-gene targets. As input into the ABC model, we used publicly available ATAC-seq and Hi-C data from murine BMDMs stimulated with IFNΓ for 2 hours (PRJNA694816) (Platanitis et al., 2022) and H3K27ac ChIP-seq and RNA-seq data from (Piccolo et al. 2017). Focusing on a permissive set of 2720 ISGs from (Piccolo et al. 2017) (FDR adjusted *p*-value <0.05, log2 fold change > 0), we identified 530 B2_Mm2 elements predicted to interact with 457 mouse ISGs (16.8% of the set of 2720 in this analysis) (Supplementary file 7). Compared with a set of human ISGs from human CD14+ monocytes (Qiao et al., 2016), 393 of these 457 (86%) genes were only ISGs in mouse, and the remaining 64 (14%) were ISGs in both mouse and human.

From this set, we decided to focus on a specific B2_Mm2 element located on Chromosome 12 within the first intron of *Dicer1*, which is an endonuclease responsible for recognizing and cleaving foreign and double stranded RNA that has been linked to innate immunity (MacKay et al. 2014; Poirier et al. 2021; Chiappinelli et al. 2012; Gurung et al. 2021). While the human ortholog *DICER1* does not show IFNΓ-inducible expression in human primary macrophages (Qiao et al., 2016) (Figure 3 —figure supplement 1, Supplementary file 1), mouse *Dicer1* shows a significant 50% upregulation in response to IFNΓ in primary mouse BMDMs (Figure 3A). This indicates that *Dicer1* is a mouse-specific ISG and likely acquired IFNΓ-inducible expression in the mouse lineage, potentially due to the co-option of the B2_Mm2 element as a species-specific IFNΓ-inducible enhancer. The intronic B2_Mm2 element (B2_Mm2.Dicer1) shows biochemical hallmarks of enhancer activity including inducible STAT1 and H3K27ac signal as well as constitutive binding by CTCF and RAD21 (Figure 3B). The element provides the only prominent nearby STAT1 binding site and is not present in rat or other mammals (Figure 3B). Therefore, we hypothesized that the B2_Mm2.Dicer1 element was co-opted as an IFNΓ-inducible enhancer of mouse *Dicer1*.”

**Author response image 1. sa2fig1:** Activity-by-Contact (ABC) analysis of B2_Mm2 enhancers and ISGs. (**A**) Proportions of ISGs (mouse-specific, human-specific, shared) predicted to be regulated by a B2_Mm2 enhancer. (**B**) Proportions of ISGs within 50 kb of a STAT1-bound B2_Mm2 element.

**Author response image 2. sa2fig2:** Distribution of Activity-by-Contact scores in mouse IFNΓ-stimulated bone-marrow derived macrophages between various TE families and mouse-specific/human-specific/shared ISGs. (**A**) B2_Mm2. (**B**) PB1D7. (**C**) ID_B1. (**D**) MT2B2. (**E**) MIR1_AMN. (**F**) MIR3.

2) The results of Serpina3g and Serpina3F gene expression in the authors' knockout cells are very interesting. However, the authors focus almost exclusively on Serpina3g and Serpina3F, which makes it difficult to understand what is happening genome wide. Are other IFNΓ-induced genes (including those not on chromosome 12) similarly affected at the level of basal or induced transcription? How many genes are different in WT versus KO cells, both at basal and induced states? Does this correlate with their CUT&TAG data shown in Figure 5? By focusing only on nearby genes (Serpina3g and Serpina3F), the authors hint that this may be a long-range regulatory effect, "potentially mediated by the CTCF binding activity of the element" that they removed. But by only focusing on two nearby IFNΓ-induced genes, their data do not rule out the (also potentially quite interesting) possibility that there may be a more indirect role for this TE site or Dicer1 in basal transcription of IFNΓ-induced genes or IFNΓ-mediated gene expression.

We agree that a more global assessment of the transcriptome-wide changes is necessary to fully understand the direct and indirect effects of the B2_Mm.Dicer1 element. However, the RNA-seq dataset for WT and KO cells generated in our initial submission was not suited for this comparison because (i) they were generated in different batches, and (ii) the WT control was a pool of WT J774 cells rather than clonal controls. Therefore, we generated new RNA-seq data where WT and KOs were treated in the same batch, using 3 additional control WT/unedited clones that were carried through the limited dilution process. For each WT (N=3) and KO (N=2) independently derived clonal line, we carried out 3 experimental replicates of untreated cells and cells treated for IFNΓ for 4 hrs, and harvested RNA for RNA-seq. These were all conducted in the same round of experiments to avoid potential batch effects.

We re-analyzed our new RNA-seq data to address the concerns raised by reviewers' comment (2) and comment (3). Consistent with our previous analysis, KO cells showed significantly reduced expression of *Dicer1* in both untreated and IFNΓ-induced conditions. We first conducted DESeq2 pairwise differential expression analysis considering all replicates, WT (N=9) vs KO (N=6) in untreated cells showed *Dicer1* downregulation with log_2_FC = -0.43, FDR adjusted *p*-value = 9×10^-4^ (see Author response image 3). In IFNΓ treated cells, the WT (N=9) vs KO (N=6) comparison showed *Dicer1* downregulation with log_2_FC = -0.80, FDR adjusted *p*-value = 3.393×10^-15^ (see Author response image 4). As before, all KO lines showed the same non-inducible expression of *Dicer1*. Within all WT clones, *Dicer1* showed significant IFNΓ-inducibility (log_2_FC = 0.35; FDR adjusted *p*-value = 1.7×10^-6^) while this inducibility was ablated in KO cells (log_2_FC = 0; FDR adjusted *p*-value = 0.98).

**Author response image 3. sa2fig3:** MA plot of all WT replicates vs all KO replicates in the untreated condition. Dicer1 is labeled and is highly expressed and modestly but significantly downregulated (log_2_FC = -0.43, FDR adjusted *p*-value = 9×10^-4)^.

**Author response image 4. sa2fig4:** MA plot of all WT replicates vs all KO replicates in the IFNΓ-treated condition. Dicer1 is labeled and is highly expressed and significantly downregulated (log_2_FC = -0.43, FDR adjusted *p*-value = 9×10^-4)^.

We next examined how *Dicer1* compares to other genes affected genome-wide by the knockout. Comparing IFNΓ-treated WT to KO cells, *Dicer1* was the 102nd most dysregulated gene (out of 3567 total genes with FDR adjusted *p*-value < 0.05). In untreated cells, *Dicer1* was ranked 1741 out of differentially expressed genes based on FDR adjusted *p*-value (out of 3992 total genes with FDR adjusted *p*-value < 0.05). In other words, in untreated KO cells, there are 1740 genes that show a more significant change than *Dicer1*. In contrast, in IFNΓ-treated KO cells, there are just 101 genes that show a more significant change than *Dicer1*. In both conditions, the other dysregulated genes are located across all chromosomes and did not show any clear functional patterns or enrichment. We did not observe correlations between these genes and our CUT&TAG data, with the caveat that our CUT&TAG was performed on just one KO clone (KO1) and the pooled WT cells. We expect that CUT&TAG data would correspond to RNA-seq data if derived from the same clone.

Based on the global distribution across the genome, the other differentially expressed genes in the KOs most likely represent off-target/stochastic changes, that are commonly seen across separate clonal isolations. We routinely use CRISPR KO and observe similar levels of off-target effects by RNA-seq even across control lines, presumably due to the process of clonal isolation. We have observed greater on-target gene expression differences when using CRISPRi (dCas9-Krab) to target enhancers in pooled cells (avoiding clonal isolation), but we decided not to use CRISPRi for this element, reasoning that the element is located within the gene intron and dCas9-Krab binding could impede *Dicer1* transcription. While it is a potentially interesting scenario that the observed dysregulation of other genes is an indirect effect of *Dicer1* downregulation, the untreated KO cells showed only a modest downregulation of *Dicer1* (log_2_FC = -0.43) which is may not sufficiently alter Dicer1 protein levels or function. The functional consequence of IFNΓ-inducible regulation on Dicer1 would be interesting to follow up in a future study.

Summary of changes

– Figure 4B-E is updated to use numbers from the NA-seq reanalysis

– Main text updated as follows

“We used RNA-seq to study the genome-wide effects of the B2_Mm2.*Dicer1* element in both knockout clones and 3 control wild-type clones which were also isolated by limiting dilution. Consistent with the RT-qPCR results, we found that *Dicer1* showed significant IFNΓ-inducible upregulation in all WT clones but that this induction was completely ablated in B2_Mm2.*Dicer1* KO clones (Figure 4B). Notably, the RNA-seq normalized count data revealed that levels of *Dicer1* were also significantly reduced in untreated knockout cells (Figure 4B, 5A). This indicates that the B2_Mm2.*Dicer1* element regulates both basal expression levels of *Dicer1* and inducible expression by IFNΓ. Focusing on the IFNΓ-treated condition, *Dicer1* was significantly downregulated in KO cells compared to WT cells cells (log2FC = -0.80, adjusted p-value = 3.393×10^-15^). Genome-wide, there were 101 genes that showed greater significance than *Dicer1* when testing for differential expression between IFNΓ-treated KO and WT cells (out of 3567 total genes with FDR adjusted *p*-value < 0.05; Supplementary file 8). The other dysregulated genes showed no discernable physical or functional pattern and also showed high variability between individual clones, consistent with clonal transcriptional variation typical of the limiting dilution process (Lee et al., 2019).”

3) The specificity of the KO effect on the Dicer1-Serpina region of chromosome 12 is not clear. Without analysis of the complete RNA-seq and CUT&RUN datasets, it is difficult to rule out a more global effect (i.e., beyond chromosome 12). If these new analyses yield evidence of specificity of the KO lines, the reviewers will be satisfied. If not, the reviewers request an additional manipulation: KO of an intron element of equivalent size to the original deletion, KO of a different B2 element – apparently there is one 2kb away, or even better, replacement of the B2 element with one that lacks cGAS motifs (though the final suggestion is likely too technically challenging). Determining whether there are changes in the basal or induced levels of Dicer1/Serpina genes in this additional line would serve as an important control for the KO experiment. Providing more data on other genes throughout the genome in WT and KO cells, which the authors have generated but do not include in the manuscript, would help distinguish between these models.

As described in our response above for Reviewer Critique #2, we have generated RNA-seq data for 3 new control lines generated by taking wild-type cells through clonal isolation. We agree with the reviewers that a "negative control KO" would be the best control, however in our past experience with CRISPR we find that most off-target/non-specific expression changes arise due to the process of clonal isolation. We also repeated the experiments and RNA-seq from our 2 KO lines such that our comparisons would not be confounded by batch effects.

Our new RNA-seq analysis of the 3 clonal WT lines and 2 KO lines confirmed that *Dicer1* is consistently IFNΓ-inducible in the controls but not inducible in the KOs, with lower basal and induced levels (updated Figure 4A-B). However, and importantly, we found that *Serpina3f* and *Serpina3g* did not show the same pattern of regulation we observed previously (significantly reduced untreated and IFNΓ-induced expression in KO cells). In the new round of RNA-seq, we found that *Serpina3f*/*g* did not show significantly or consistently reduced expression in untreated KO cells (Supplementary file 8). These results conflict with our previous claim that *Serpina3f* and *Serpina3g* are regulated by the B2_Mm2.Dicer1 element, and instead suggest that our previous observation was an artifact of clonal variation and the extremely strong (log2fc > 10) IFNΓ-inducibility of *Serpina3g* within each genotype (in WT, inducible log_2_FC = 11.3, FDR adjusted *p*-value = 4.27×10^-183^; in KO, inducible log_2_FC 10.4; FDR adjusted *p*-value ≈ 0) and *Serpina3f* (WT log_2_FC = 10.4, FDR adjusted *p*-value = 4.8E×10^-99^; KO log_2_FC 10.3, FDR adjusted *p*-value = 9.3×10^-100^). It is likely that our previous observation was due to the fact that our KO1/2 RNA-seq datasets were generated in a separate experimental batch than the WT control, which was a pooled set of cells. Furthermore, our chromatin interaction analysis (described in response to Reviewer Critique #1) predicts that the B2_Mm2.Dicer1 element targets *Dicer1*, but does not target the *Serpina* genes. Taken together, our new controls support a specific regulatory effect of the element on *Dicer1* but not the *Serpin* genes (as previously claimed). As such, we have removed our claims that *Serpin* genes are affected by the B2_Mm2.Dicer1 knockout from the main text. While our re-analysis indicates that the B2_Mm2.Dicer1 element does not have long-range enhancer activity as we previously concluded, this does not affect the major claims of our study. While the long-range distal enhancer activity was potentially an interesting observation, the fact that B2_Mm2.Dicer1 has a clear and reproducible regulatory effect on the more proximal gene *Dicer1* is consistent with our central claim that B2_Mm2 elements act as IFNΓ-inducible enhancers.

The following paragraph was removed from the main text:

Our RNA-seq analysis revealed that deletion of B2_Mm2.Dicer1 also had a significant repressive effect on the IFNΓ-inducible expression of *Serpina* genes. In mouse, *Serpina* is a family of inflammatory genes represented by a large cluster of 14 murine paralogs, located 330 – 1000 kb from B2_Mm2.Dicer1. In particular, we found that *Serpina3f* and *Serpina3g* (approximately 500 kb from B2_Mm2.Dicer1) show a 1-2 orders of magnitude reduction in basal and induced gene expression in knockout cells compared to WT cells (Figure 4E-F, 5B). The relatively long range of this regulatory activity is not uncommon in vertebrate genomes (Schoenfelder and Fraser, 2019) and is potentially mediated by the CTCF binding activity of the element.

4) There are high levels of POLR2A occupancy at the B2_Mm2.Dicer1 element in induced WT cells. Could this be a Pol2 pause site? Could deletion of this element lead to a change in Pol2 occupancy and change Dicer1 expression independent of enhancer activity? To probe such questions, the reviewers requested that the authors directly test the possibility that the intronic B2 element actually acts as a regulator of splicing or transcriptional elongation. Careful analysis of the Dicer1-mapping reads from the RNA-seq data – or RT-qPCR – could resolve this concern.

We agree that it would be interesting if the B2_Mm2.Dicer1 altered *Dicer1* transcription separately from its function as an inducible enhancer, and thank the reviewer for the suggestion. As shown in Figure 5, deletion of the B2_Mm2.Dicer1 element does remove Pol2 occupancy over the element and reduces overall signal over the locus, consistent with enhancer activity but not mutually exclusive with pausing activity. If the intronic B2_Mm2.Dicer1 element acts as a regulator of splicing, we would expect changes in isoform expression. As suggested, we manually inspected aligned RNA-seq reads from all WT and KO untreated/treated samples. Qualitatively, we observed no evidence of differential splicing or exon skipping when looking at aligned fragments from WT J774 and B2_Mm2.Dicer1 KO J774 cells (see below).

To more formally analyze this possibility with the data available, we conducted transcript assembly on all the samples and tested for differential isoform usage focusing on *Dicer1* transcripts. As suggested by the reviewers, if the B2_Mm2.Dicer1 element acts as a pause site, it may alter splice site or polyadenylation site usage and cause measurable differential expression of specific *Dicer1* splice variants in KO cells. To test this, we ran Stringtie v2 to conduct transcript assembly using the RNA-seq reads from each WT and KO individual sample, then used the ‘--merge’ flag to generate a merged gene annotation file representing isoforms from all samples. In parallel we also generated merged GTF files created using alignment files only from WT or B2_Mm2.Dicer1 KO samples. All exons identified in the fully merged GTF file are represented in the WT- and KO-specific GTF files. At the transcript level, all *Dicer1* isoforms in the fully merged GTF file are also represented in the WT- and KO-specific GTF file. Together, this indicates that there are no isoforms that are unique to either KO and WT cells (see Author response image 5).

**Author response image 5. sa2fig5:** UCSC Snapshot of the *Dicer1* locus in mouse, with custom Stringtie transcript-level annotation (top three tracks), GENCODE gene-level annotation (fourth track), and alignments from clonal WT (fifth track) and B2_Mm2. Dicer1 KO (sixth track) J774.A1 cells.

To quantify differences in isoform expression, we used DESeq2 to measure differences in isoform expression across WT and B2_Mm2.Dicer1 KO samples using the merged Stringtie isoform information as input (gene-level analysis was used for the manuscript). This revealed several transcripts that all show similar trends to the gene-level analysis (see Supplementary file 9), where expression is reduced in both basal and IFN-treated conditions in the KO, and inducible expression is lost. If the element acted as a pause site affecting elongation, we would have expected unrestricted Pol2 elongation in both conditions and therefore greater expression levels in the KO in either or both basal and treated conditions. However, we only observed reduced expression of transcripts in all conditions in the KO, which is most consistent with activity as an enhancer.

We have uploaded these alignments and accompanying Stringtie annotation files to an updated UCSC Genome Browser session (https://genome.ucsc.edu/s/coke6162/B2_SINE_enhancers_Horton_et_al). These analyses did not provide strong evidence supporting splicing changes driven by the KO, although a major caveat is that the isoforms were analyzed using short-read sequencing and therefore may not be accurately assembled or quantified, and high-coverage long-read sequencing may be required for more accurate isoform quantification.

In addition to analyzing our RNA-seq data as suggested by the reviewers, we also looked for evidence of ChIP-seq binding at the B2 element by pausing associated factors including NELF and CDK9 using published datasets atalogued by the mouse Cistrome ChIP-seq binding site database. We did not find evidence for binding by any canonical pausing factors at either the B2 element or the *Dicer1* promoter.

B2 element: http://dbtoolkit.cistrome.org/?specie=mm10&factor=tf&interval=chr12%3A104742467-104742646#plot_result

Dicer1 promoter: http://dbtoolkit.cistrome.org/?specie=mm10&factor=tf&interval=chr12%3A104%2C751%2C526-104%2C751%2 C993#plot_result

We also downloaded signal coverage tracks from a recently published *NELF* KO PRO-seq and NELF/CDK9 ChIP-seq dataset from mouse macrophages either untreated or stimulated with LPS from (Yu et al., 2020) and inspected the B2_Mm2.Dicer1 region (see Author response image 6). We did not find any evidence for activity over the B2 element in this dataset in either untreated or treated conditions, with the caveat that the study used LPS treatment instead of IFNΓ.

**Author response image 6. sa2fig6:** IGV Snapshot of the *Dicer1* locus in mouse, with PRO-Seq of WT and NELF KO (pause-release deficient) mouse bone-marrow derived macrophages from (Yu et al., 2020).

Overall, these analyses do not provide strong support for the element acting as a pause site, but do not completely rule out the possibility. For example, it is possible that the element serves as a pause site in vitro, and is only released by NELF and other factors specifically under IFNΓ conditions, driving inducible elongation of *Dicer1* in WT cells. However, the fact that KO of the element reduced basal transcript levels of *Dicer1* more strongly supports enhancer activity. Direct ChIP-weq investigation of NELF/P-Tefb (CDK9/Cyclin T1) in our IFNΓ-treated WT and KO cells, as well as use of nascent transcription (PRO-seq), are necessary to directly investigate Pol2 pausing. However, given that did we did not observe any clear differences at the transcript level, we feel that our findings are most consistent with the element acting as a typical inducible cis-regulatory enhancer, and investigation of pausing would be outside the scope of the study.

We have added our analysis of isoform-level changes to the main text as follows:

“Given the B2_Mm2.Dicer1 element is bound by POLR2A in WT cells, we examined whether the B2_Mm2.Dicer1 element may alter transcription by affecting usage of different splice sites or polyadenylation sites, which would be consistent with pause site activity. We examined transcript isoform-level expression changes in KO cells in both untreated and treated conditions and found multiple transcripts that showed the same trend as the gene-level analysis, where most expressed transcripts are downregulated in both basal and IFNΓ-treated conditions and show lack of inducibility in KO cells (Supplementary file 9). These findings are consistent with the element acting primarily as an IFNΓ-inducible enhancer without any major effect on alternative splicing.”

5) Figure 4F – The authors claim that "deletion of B2_Mm2.Dicer1 also has a significant repressive effect on the IFNΓ-inducible expression of Serpina genes." However, the basal levels of Serpina3f/Serpina3g are significantly reduced upon this deletion compared to WT. Furthermore, expression of Serpina genes in the KO cell lines significantly increase upon IFNΓ stimulation, suggesting that they still show inducible expression despite the B2_Mm2.Dicer1 deletion. The authors should compare the magnitude of induction before and after stimulation between the WT and KO cell lines to determine if there is indeed a repressive effect on inducible expression of Serpina genes.

We thank the reviewers for encouraging us to further validate the regulatory effect on the Serpina genes. In our original study, the WT and KO cell lines were treated and sequenced in different batches, which confounded our comparison. As we describe in our answer to reviewer comment (3), we have re-generated and re-analyzed RNA-seq data from multiple WT and KO clonal lines in both untreated/treated conditions, with all data generated in the same experimental batch. This reanalysis confirmed the specificity of the enhancer's inducible regulation of *Dicer1*, but revealed no consistent effect on *Serpina3f*/*a3g*. This suggests that our previous finding of *SerpinA3g* was an artifact likely due to clonal variation and/or batch effect, as our reanalysis using new control clones that were isolated by limited dilution no longer showed a consistent effect on the *Serpina* genes. Therefore we have removed our findings on the enhancer's long-range regulation of *Serpina3g* and focus on the effect of enhancer activity on *Dicer1*. We do not believe this alters our central claims, which is that B2_Mm2 elements are an abundant source of IFNΓ-inducible enhancers, which typically regulate proximal genes within 50 kb of the gene.

[Editors' note: further revisions were suggested prior to acceptance, as described below.]

The reviewers appreciated your many additional experiments/analyses as well as the edits to the text. The manuscript has been improved substantially but there are some remaining issues that need to be addressed.1. All three reviewers felt that the response to Critique#1 was insufficient. There are several places in the manuscript referring to the significance of the Mouse B2 SINE elements contributing to lineage-specific immune response. For example, the introduction highlights that "a key open question is whether the co-option of TEs as immune regulatory elements is evolutionarily widespread as a mechanism driving divergence of innate immune responses."The reviewers (or at least two of them) did not expect to see additional KO experiments but did expect the authors to highlight a handful of other examples where STAT1 binding to a proximal B2 SINE element is found in a mouse-specific ISG only. If no other such examples are available to highlight, then it's currently difficult to discern how generalizable this relationship is between rodent-specific B2 SINE elements and mouse-specific ISGs.

We thank the reviewers for the suggestion to highlight examples of species-specific mouse ISGs predicted to be regulated by B2_Mm2 elements. We identified 393 B2_Mm2 elements predicted regulate mouse-specific ISGs based on chromatin state and interaction (ABC model (Fulco et al., 2019)). We have added the information on these candidate elements and their target gene as a new sheet in Supplementary File 7. We also highlighted 5 additional examples in new Figure 3 supplements (Figure 3 —figure supplement 1, Figure Supplement 2), based on high ABC predicted interaction scores and published associations with immunity. We also conducted GO enrichment on these 393 genes and identified significant enrichment of terms related to immune responses, which we have also reported in a new sheet in Supplementary File 7. We have added the following paragraph to the main text:

“The 393 mouse-specific ISGs predicted to be regulated by B2_Mm2 elements were significantly enriched for multiple immune related functions (GO:0002376, adjusted *p*-value = 6.023×10^-9^; Supplementary file 7). We identified multiple examples of predicted B2_Mm2 target genes with established immune functions that showed mouse-specific IFNΓ-inducible expression (Figure 3 Figure Supplement 1, Figure Supplement 2), including dicer 1 ribonuclease III (*Dicer1*) (Poirier et al., 2021) (Figure 3A), SET domain containing 6, protein lysine methyltransferase (SETD6) (Levy et al., 2011), DOT1-like histone lysine methyltransferase (Kealy et al., 2020), fumarate hydratase 1 (Fh1) (Zecchini et al., 2023), heat shock protein family A (Hsp70) member 1B (*Hspa1b*) (Jolesch et al., 2012), NFKB inhibitor δ (*Nfkbid*) (Souza et al., 2021).”

Furthermore, the authors highlight a limitation of the current human cell dataset – specifically the limitation of sampling gene expression only 24 hours after interferon treatment (new paragraph starting line 310) in human cells. However, in line 232, the same dataset is used to support the earlier statement that "the human ortholog DICER1 does not show IFNΓ-inducible expression in human primary macrophages (Qiao et al. 2016)…mouse DICER1 shows a significant 50% upregulation in response to IFNΓ in primary mouse BMDMs…."This earlier statement is substantially weakened by the final Results paragraph. This issue needs to be resolved to support the major claim of the paper that a lineage-specific TE is responsible for a lineage-specific immune response, echoed in the abstract: "B2 elements…exemplifies how lineage-specific TEs can facilitate evolutionary divergence of innate immune regulatory networks."

During the first revision, we found that the human DICER1 locus harbors STAT1 binding sites, despite not showing IFNΓ-inducible expression in primary monocytes (Qiao et al. 2013, 2016). While it is possible the STAT1 binding sites are nonfunctional, this prompted us to analyze RNA-seq data of IFNΓ-stimulated monocytes from another group (McCann et al. 2022) and saw that *DICER1* did show IFNΓ-inducible expression in this other dataset. Thus, the evidence for inducible expression of human *DICER1* was inconsistent but we felt it was important to include this additional analysis. There are several factors that could drive this observation, including the fact that these studies were conducted by different groups from cells from different donors. We recognize that this observation weakens our original claim stating that B2 elements drive lineage-specific expression and "evolutionary divergence of innate immune regulatory networks." However, our finding is consistent with conclusions from other groups (Choudhary et al., 2020; Sundaram et al., 2014) that lineage-specific TEs can contribute to regulatory turnover of transcription factor binding sites and/or convergent evolution.

We emphasize that B2 elements are still driving lineage-specific evolution of the cis-regulatory architecture underlying the IFNΓ regulatory network. But, at the gene expression level, different B2 elements may drive lineage-specific divergence or maintenance/turnover (replacement of existing sites) (Dermitzakis and Clark, 2002; Moses et al., 2006) or convergent regulatory evolution, which are important mechanisms of regulatory evolution (Long et al., 2016). Distinguishing the last two possibilities would require examining *Dicer1* inducibility by IFNΓ in monocytes from other species, which would be interesting but outside the scope of this study. Furthermore, given that we identified 393 other B2 elements predicted to act as species-specific enhancers (based on our comparison of the Piccolo et al. and Qiao et al. 2016 datasets), we expect that a substantial fraction are driving lineage-specific divergence of inducible gene expression.

We have rewritten the final results paragraph to clarify that this idea:

“Considering that the B2_Mm2 enhancer is specific to rodents, we examined the regulatory landscape of the human *DICER1* locus. Our analysis of RNA-seq data from human primary monocytes treated with IFNΓ for 24 hrs (Qiao et al., 2016) indicated that human *DICER1* expression is not induced by IFNΓ (Supplementary file 1, Figure 3 —figure supplement 3). However, ChIP-seq data from IFN-treated monocytes from the same group (Qiao et al., 2013) showed multiple inducible STAT1 binding sites within the human *DICER1* locus, including one originating from a primate-specific TE (LTR27) (Figure 3—figure supplement 3). Although these binding sites do not correlate with inducible *DICER1* expression in the matched RNA-seq dataset, they suggest human *DICER1* may be inducible under different conditions. An analysis of an independent dataset generated from another donor (McCann et al., 2022) supported the inducible expression of *DICER1* (log_2_FC = 0.91 & FDR adjusted *p*-value = 3.0×10^-4^) (Supplementary file 1). Thus, while the evidence for inducible human *DICER1* expression is inconsistent, our analyses indicate that human *DICER1* has independently evolved primate-specific binding STAT1 binding sites, which may also confer inducible regulation.”

We added the following paragraph to the discussion:

“We identified hundreds of B2_Mm2-derived enhancers that are predicted to regulate genes displaying IFNΓ-inducible expression in mouse cells but not human cells, which supports their role in facilitating lineage-specific evolution of the IFNΓ-inducible regulatory network. However, we also identified a subset of target genes that show inducible expression in both species, suggesting independent evolution or turnover of regulatory elements that could serve similar regulatory functions as the B2_Mm2 enhancers. For instance, we identified an intronic STAT1 binding site derived from a primate-specific TE in the human *DICER1* locus. While we did not uncover consistent evidence supporting IFNΓ-inducible regulation of human *DICER1*, these observations align with the concept of convergent regulatory evolution, in which similar expression patterns are perpetuated by the co-option of lineage-specific TEs (Choudhary et al., 2020; Sundaram et al., 2014). Therefore, as B2_Mm2 elements shaped the evolution of rodent immune regulatory networks, individual co-option events may have mediated either divergence or preservation of gene expression patterns.”

We have also edited the last sentence of the abstract as follows to be consistent with our Dicer1 findings:

“Regulatory networks underlying innate immunity continually face selective pressures to adapt to new and evolving pathogens. Transposable elements (TEs) can affect immune gene expression as a source of inducible regulatory elements, but the significance of these elements in facilitating evolutionary diversification of innate immunity remains largely unexplored. Here, we investigated the mouse epigenomic response to type II interferon (IFN) signaling and discovered that elements from a subfamily of B2 SINE (B2_Mm2) contain STAT1 binding sites and function as IFN-inducible enhancers. CRISPR deletion experiments in mouse cells demonstrated that a B2_Mm2 element has been co-opted as an enhancer driving IFN-inducible expression of *Dicer1*. The rodent-specific B2 SINE family is highly abundant in the mouse genome and elements have been previously characterized to exhibit promoter, insulator, and non-coding RNA activity. Our work establishes a new role for B2 elements as inducible enhancer elements that influence mouse immunity, and exemplifies how lineage-specific TEs can facilitate evolutionary turnover and divergence of innate immune regulatory networks.”

2. In the response to reviewer comments under critique#2, it was stated that "…the other differentially expressed genes in the KOs most likely represent off-target/stochastic changes, that are commonly seen across separate clonal isolations." And as communicated in the manuscript: Line 288: "The other dysregulated genes showed no discernable physical or functional pattern and also showed high variability between individual clones…consistent with intrinsic clonal transcriptional variation…" Whether this is true for all 101 genes should be clarified. It seems surprising given that at least some genes should be indirect targets of Dicer1. Finally, does Dicer1 show less variability than these other genes? If so, that should be stated.

We agree with the reviewers that some of these genes are potentially indirect targets of *Dicer1* which would be interesting. Given that the role of *Dicer1* is mRNA degradation, we expect any indirect targets to increase in expression after *Dicer1* knockout, and 35/101 of the genes were. We have included these numbers in the revision. However, we emphasize that CRISPR KO experiments often result in large unintended transcriptional changes and we cannot exclude the possibility that these changes are due to clonal variation or off-target effects.

We have added the following paragraph to the results

“Genome-wide, there were 101 genes that showed greater significance than *Dicer1* when testing for differential expression between IFNΓ-treated KO and WT cells (out of 3567 genes with FDR-adjusted *p*-value < 0.05; Supplementary file 8). Out of these 101 genes, 24 showed higher variability than *Dicer1* (based on DESeq2 log2foldchange standard error) between individual clones (Figure 3 – figure supplement 5B-E), consistent with intrinsic clonal transcriptional variation revealed by the limiting dilution and/or CRISPR editing process (Nahmad et al., 2022; Westermann et al., 2022). 35 of these genes showed upregulation in the KO cells, suggesting that they could be silencing targets of *Dicer1* that become upregulated upon *Dicer1* downregulation. However, given the relatively modest effect of *Dicer1* especially in the untreated condition, further experiments would be necessary to establish these genes as targets of *Dicer1*.”

3. Finally, one reviewer was concerned that the intronic location of B2 SINE means that the authors cannot delineate whether the element acts as an enhancer or instead a regulator of transcriptional elongation.

We agree that we cannot exclude the possibility that the element acts as a regulator of transcriptional elongation due to its position in the intron. We have added this caveat in our results at the end of the paragraph describing this analysis:

“However, further experiments such as CDK9 inhibition and profiling of nascent transcription in stimulated conditions (Gressel et al., 2017; Laitem et al., 2015) are necessary to establish whether the element affects transcriptional elongation.”

References

Choudhary MN, Friedman RZ, Wang JT, Jang HS, Zhuo X, Wang T. 2020. Co-opted transposons help perpetuate conserved higher-order chromosomal structures. *Genome Biol* 21:16.

Dermitzakis ET, Clark AG. 2002. Evolution of transcription factor binding sites in Mammalian gene regulatory regions: conservation and turnover. *Mol Biol Evol* 19:1114–1121.

Fulco CP, Nasser J, Jones TR, Munson G, Bergman DT, Subramanian V, Grossman SR, Anyoha R, Doughty BR, Patwardhan TA, Nguyen TH, Kane M, Perez EM, Durand NC, Lareau CA, Stamenova EK, Aiden EL, Lander ES, Engreitz JM. 2019. Activity-by-contact model of enhancer-promoter regulation from thousands of CRISPR perturbations. *Nat Genet* 51:1664–1669.

Gressel S, Schwalb B, Decker TM, Qin W, Leonhardt H, Eick D, Cramer P. 2017. CDK9-dependent RNA polymerase II pausing controls transcription initiation. *ELife* 6. doi:10.7554/*eLife*.29736

Jolesch A, Elmer K, Bendz H, Issels RD, Noessner E. 2012. Hsp70, a messenger from hyperthermia for the immune system. *Eur J Cell Biol* 91:48–52.

Kealy L, Di Pietro A, Hailes L, Scheer S, Dalit L, Groom JR, Zaph C, Good-Jacobson KL. 2020. The Histone Methyltransferase DOT1L Is Essential for Humoral Immune Responses. *Cell Rep* 33:108504.

Laitem C, Zaborowska J, Isa NF, Kufs J, Dienstbier M, Murphy S. 2015. CDK9 inhibitors define elongation checkpoints at both ends of RNA polymerase II-transcribed genes. *Nat Struct Mol Biol* 22:396–403.

Levy D, Kuo AJ, Chang Y, Schaefer U, Kitson C, Cheung P, Espejo A, Zee BM, Liu CL, Tangsombatvisit S, Tennen RI, Kuo AY, Tanjing S, Cheung R, Chua KF, Utz PJ, Shi X, Prinjha RK, Lee K, Garcia BA, Bedford MT, Tarakhovsky A, Cheng X, Gozani O. 2011. Lysine methylation of the NF-κB subunit RelA by SETD6 couples activity of the histone methyltransferase GLP at chromatin to tonic repression of NF-κB signaling. *Nat Immunol* 12:29–36.

Long HK, Prescott SL, Wysocka J. 2016. Ever-Changing Landscapes: Transcriptional Enhancers in Development and Evolution. *Cell* 167:1170–1187.

McCann KJ, Christensen SM, Colby DH, McGuire PJ, Myles IA, Zerbe CS, Dalgard CL, Sukumar G, Leonard WJ, McCormick BA, Holland SM. 2022. IFNγ regulates NAD+ metabolism to promote the respiratory burst in human monocytes. *Blood Adv* 6:3821–3834.

Moses AM, Pollard DA, Nix DA, Iyer VN, Li X-Y, Biggin MD, Eisen MB. 2006. Large-scale turnover of functional transcription factor binding sites in *Drosophila*. *PLoS Comput Biol* 2:e130.

Nahmad AD, Reuveni E, Goldschmidt E, Tenne T, Liberman M, Horovitz-Fried M, Khosravi R, Kobo H, Reinstein E, Madi A, Ben-David U, Barzel A. 2022. Frequent aneuploidy in primary human T cells after CRISPR-Cas9 cleavage. *Nat Biotechnol* 40:1807–1813.

Poirier EZ, Buck MD, Chakravarty P, Carvalho J, Frederico B, Cardoso A, Healy L, Ulferts R, Beale R, Reis e Sousa C. 2021. An isoform of Dicer protects mammalian stem cells against multiple RNA viruses. *Science* 373:231–236.

Qiao Y, Giannopoulou EG, Chan CH, Park S-H, Gong S, Chen J, Hu X, Elemento O, Ivashkiv LB. 2013. Synergistic activation of inflammatory cytokine genes by interferon-γ-induced chromatin remodeling and toll-like receptor signaling. *Immunity* 39:454–469.

Qiao Y, Kang K, Giannopoulou E, Fang C, Ivashkiv LB. 2016. IFN-γ Induces Histone 3 Lysine 27 Trimethylation in a Small Subset of Promoters to Stably Silence Gene Expression in Human Macrophages. *Cell Rep* 16:3121–3129.

Souza SP, Splitt SD, Sànchez-Arcila JC, Alvarez JA, Wilson JN, Wizzard S, Luo Z, Baumgarth N, Jensen KDC. 2021. Genetic mapping reveals Nfkbid as a central regulator of humoral immunity to *Toxoplasma gondii*. *PLoS Pathog* 17:e1010081.

Sundaram V, Cheng Y, Ma Z, Li D, Xing X, Edge P, Snyder MP, Wang T. 2014. Widespread contribution of transposable elements to the innovation of gene regulatory networks. *Genome Res* 24:1963–1976.

Westermann L, Li Y, Göcmen B, Niedermoser M, Rhein K, Jahn J, Cascante I, Schöler F, Moser N, Neubauer B, Hofherr A, Behrens YL, Göhring G, Köttgen A, Köttgen M, Busch T. 2022. Wildtype heterogeneity contributes to clonal variability in genome edited cells. *Sci Rep* 12:18211.

Zecchini V, Paupe V, Herranz-Montoya I, Janssen J, Wortel IMN, Morris JL, Ferguson A, Chowdury SR, Segarra-Mondejar M, Costa ASH, Pereira GC, Tronci L, Young T, Nikitopoulou E, Yang M, Bihary D, Caicci

F, Nagashima S, Speed A, Bokea K, Baig Z, Samarajiwa S, Tran M, Mitchell T, Johnson M, Prudent J, Frezza C. 2023. Fumarate induces vesicular release of mtDNA to drive innate immunity. *Nature* 615:499–506.